# BALANCING DOMAIN-INVARIANT AND DOMAIN-SPECIFIC KNOWLEDGE FOR DOMAIN GENERALIZATION WITH ONLINE KNOWLEDGE DISTILLATION

## ABSTRACT

Deep learning models often experience performance degradation when the distribution of testing data differs from that of training data. Domain generalization addresses this problem by leveraging knowledge from multiple source domains to enhance model generalizability. Recent studies have shown that distilling knowledge from large pretrained models effectively improves a model's ability to generalize to unseen domains. However, current knowledge distillation-based domain generalization approaches overlook the importance of domain-specific knowledge and rely on a two-stage training process, which limits the effectiveness of knowledge transfer. To overcome these limitations, we propose the Balanced Online knowLedge Distillation (BOLD) framework for domain generalization. BOLD employs a multi-domain expert teacher model, with each expert specializing in specific source domains to preserve domain-specific knowledge. This approach enables the student to distil both domain-invariant and domain-specific knowledge from the teacher. Additionally, BOLD adopts an online knowledge distillation strategy where the teacher and students learn simultaneously, allowing the teacher to adapt based on the student's feedback, thereby enhancing knowledge transfer and improving the student's generalizability. Extensive experiments conducted with state-of-the-art baselines on seven domain generalization benchmarks demonstrate the effectiveness of the BOLD framework. We also provide a theoretical analysis that underscores the effectiveness of domain-specific knowledge and the online knowledge distillation strategy in domain generalization. The code is available at `https://anonymous.4open.science/r/BOKD-ICLR-3FF8/README.md`.

## 1 INTRODUCTION

The success of deep neural networks largely depends on the assumption that training (source domain) and testing (target domain) data are independently and identically distributed (i.i.d.). However, this assumption is often violated in real-world scenarios due to discrepancies between training and testing data, known as the domain shift problem, leading to significant performance degradation (Wang et al., 2022). To address this problem, Domain adaptation has been explored to transfer knowledge from source to target domains (Pan & Yang, 2009). Unsupervised domain adaptation, in particular, leverages unlabelled data from target domains, thereby eliminating the need for target domain annotations (Xu et al., 2019). Despite their effectiveness, unsupervised domain adaptation methods necessitate data collection and model tuning for each target domain, making them impractical in many situations (Yue et al., 2019). Consequently, domain generalization has emerged as a prominent alternative. Domain generalization aims to learn a universal representation from multiple labelled source domains, enabling robust generalization to unseen domains (Wang et al., 2022). Existing approaches typically fall into three categories: data augmentation (Zhou et al., 2020), domain-invariant representation (Wang et al., 2022), and specialized training strategies (Zhao et al., 2024).

Knowledge distillation has recently shown promise in domain generalization (Wang et al., 2021; Huang et al., 2023). Unlike classic domain generalization methods that train models directly using one-hot ground truth labels, knowledge distillation-based approaches facilitate knowledge transfer from a complex teacher model to a simple student model. This process reduces the learning

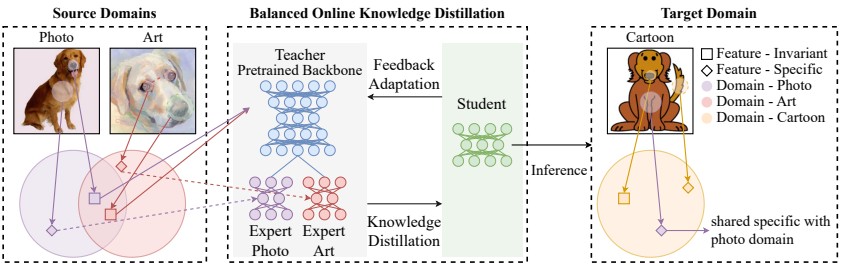

Figure 1: **Illustration of the significance of domain-specific knowledge in domain generalization.** Source domains contain both domain-invariant features, which are common across all domains, and domain-specific features, which are unique to individual domains, *e.g.* edge features from the Art domain and colour features from the Photo domain. The target domain (Cartoon) shares not only domain-invariant features with all source domains but also domain-specific features with some domains. Therefore, in addition to domain-invariant features, domain-specific features may also enhance the model's generalization performance.

complexity for the student while enabling it to acquire effective representations for domain generalization (Gou et al., 2021). However, current knowledge distillation-based methods for domain generalization have two key limitations. First, most existing approaches prioritize distilling domain-invariant knowledge, assuming that domain-specific knowledge impedes generalization (Lee et al., 2022). This assumption may not always hold, as collective comprehensive data from diverse domains is challenging, leading to domain-invariant knowledge derived from these source domains may not always generalize well to unseen target domains (Zhang et al., 2023b). As illustrated in Figure 1, target domains may share characteristics with certain source domains, suggesting that domain-specific knowledge from these sources could enhance generalization performance. To enable the student to distil domain-specific knowledge, the teacher model must first acquire this knowledge, which leads to the second limitation: most existing methods employ an offline distillation strategy, requiring a separate training phase where the teacher model is trained before guiding the student (Huang et al., 2023). Since the teacher model is fixed after its initial training, it will not adapt to the student's evolving needs during distillation, potentially resulting in ineffective knowledge transfer and limiting the student's ability to generalize.

To enable the student model to distil both domain-invariant and domain-specific knowledge while allowing the teacher to adapt to student feedback, we propose the Balanced Online knowLedge Distillation (BOLD) framework for domain generalization. BOLD leverages adapter techniques to construct a multi-domain expert teacher model. Specifically, BOLD integrates multiple adapters into a pretrained backbone model, with each adapter specializing in domain-specific knowledge for a particular domain. This design allows the student model to distil domain-invariant knowledge from the pretrained backbone and domain-specific knowledge from the corresponding domain expert adapter. Furthermore, BOLD employs an online knowledge distillation strategy, where the domain expert adapters in the teacher model are trained concurrently with the student model. During the distillation process, the domain expert adapters also minimize the discrepancy between their output and the student's output. This online approach enables the domain expert adapters to adapt to student feedback throughout training, supporting an end-to-end training scheme.

**Contribution**. Our contributions are summarized as follows. (1) We demonstrate that adapting the teacher model based on feedback from the student through online knowledge distillation improves knowledge transfer, thereby enhancing the student model's generalization capability. To the best of our knowledge, our work is the first investigation into the effectiveness of online knowledge distillation for domain generalization. (2) We show that distilling both domain-invariant and domain-specific knowledge, rather than focusing solely on domain-invariant knowledge, enhances model generalizability. (3) We provide a theoretical analysis demonstrating the effectiveness of domain-specific knowledge when the target domain shares characteristics with source domains, as well as the benefits of the online knowledge distillation strategy for domain generalization. Extensive experiments against state-of-the-art baselines across seven domain generalization benchmarks confirm the effectiveness of the BOLD framework.

## 2 RELATED WORK

**Domain Shift** refers to the degradation in performance caused by discrepancies between the source (training) and target (testing) domains (Pan & Yang, 2009). Domain adaptation has been proposed to address this issue by aligning the marginal (Baktashmotlagh et al., 2013) or conditional (Luo et al., 2020) distributions of the source and target domains or by fine-tuning models trained on source domains to adapt to the target domain (Long et al., 2015). To reduce the cost associated with annotating target domain data, domain adaptation has been explored in semi-supervised (Saito et al., 2019) and unsupervised (Long et al., 2017) scenarios, utilizing partially labelled or unlabelled target domain data during training. However, these methods still rely on pre-collected target domain data, which presents a practical limitation, as obtaining such data is not always feasible (Yue et al., 2019). This limitation highlights the need for approaches that can generalize to unseen domains without requiring target domain data collection in advance (Wang et al., 2022).

**Domain Generalization** was first introduced by Blanchard et al. (2011) and later formalized by Muandet et al. (2013). Existing domain generalization approaches primarily fall into three categories: data augmentation (Zhou et al., 2020), domain-invariant representation learning (Wang et al., 2022), and specialized learning strategies (Zhao et al., 2024). Recently, knowledge distillation has attracted attention in the context of domain generalization. Wang et al. (2021) first proposed a gradient regularization method to regularize the domain-invariant knowledge distilled from the teacher model. Lee et al. (2022) introduced a self-distillation framework where a group of students collectively form a teacher, with each student distilling domain-invariant knowledge from the ensemble teacher. Huang et al. (2023) proposed leveraging the text encoder of a Vision-Language model to distil domain-invariant knowledge. Zhang et al. (2023b) suggested distilling domain-aware knowledge from a large pre-trained teacher model. Most existing methods focus exclusively on distilling domain-invariant knowledge, overlooking the significance of domain-specific knowledge in domain generalization (Chang et al., 2019; Seo et al., 2020; Bui et al., 2021). Additionally, these methods typically employ an offline knowledge distillation strategy, where the teacher model remains fixed after initial training. In contrast, our framework distills both domain-invariant and domain-specific knowledge using an online knowledge distillation strategy, allowing the teacher to adapt based on feedback from the student.

**Knowledge Distillation** was initially developed for model compression, with the goal of making the output of a smaller student model similar to that of a larger, existing teacher model (Hinton et al., 2014). Luo et al. (2016) demonstrated that training a student model using knowledge from a teacher via knowledge distillation can lead to better performance than direct training with one-hot ground truth labels. In reinforcement learning, knowledge distillation, also known as policy distillation (Ashok et al., 2018; Liu et al., 2020; Xu et al., 2020), is employed for model compression, accelerating network training, and merging multiple agent models. Knowledge distillation methods can be categorized into offline and online approaches, depending on whether the teacher model is updated concurrently with the student model Gou et al. (2021). In offline distillation, knowledge is transferred from a pre-trained teacher to a student, typically following a two-stage training process (Zagoruyko & Komodakis, 2017; Mirzadeh et al., 2020; Li et al., 2020). Conversely, online distillation allows for the simultaneous updating of both teacher and student models and supports an end-to-end trainable knowledge distillation framework (Anil et al., 2018; Zhang et al., 2018; Chen et al., 2020; Wu & Gong, 2021). While offline knowledge distillation has proven effective in domain generalization (Wang et al., 2021; Lee et al., 2022; Huang et al., 2023), the potential of online knowledge distillation remains unexplored. To our knowledge, this work is the first to explore and theoretically analyze how online knowledge distillation enhances domain generalization.

## 3 METHODOLOGY

In this section, we first provide the preliminaries on domain generalization and knowledge distillation. We then outline the details of BOLD in two parts. First, we describe how the teacher model learns domain-specific knowledge and how the student distils domain-invariant and domain-specific knowledge from the teacher. Then, we explain how the teacher model adapts based on feedback from the student. Finally, we offer a theoretical analysis to illustrate the importance of domain-specific knowledge for domain generalization and the advantages of using an online distillation strategy. Figure 2 provides an overview of BOLD, and the algorithm is detailed in Algorithm 1.

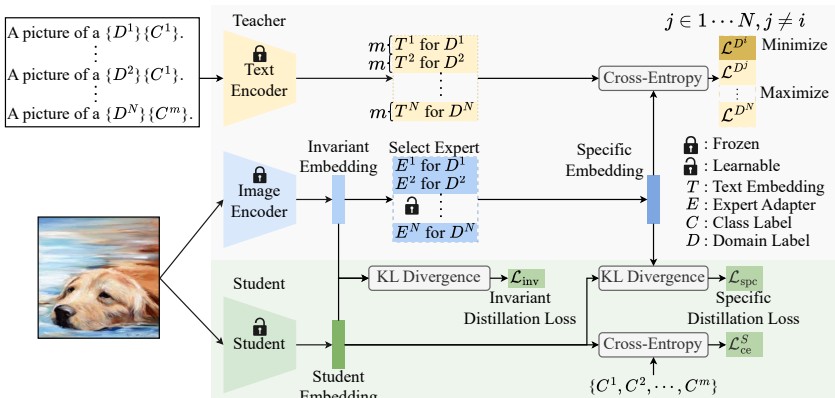

Figure 2: Overview of BOLD. BOLD employs teacher-student architecture, where the teacher model is based on Contrastive Language-Image Pretraining and consists of both an image encoder and a text encoder. The image encoder is augmented with multiple domain expert adapters to retain domain-specific knowledge for each source domain. The student distils domain-invariant knowledge by minimizing its output against the invariant embedding produced by the image encoder (invariant distillation loss) and distils domain-specific knowledge by minimizing its output against the specific embedding produced by the adapter (specific distillation loss). The domain expert adapters capture domain-specific knowledge by minimizing the image-to-text loss for the matched domain and maximizing it for unmatched domains. Additionally, they minimize specific distillation loss to incorporate feedback from the student, thereby enhancing the effectiveness of knowledge transfer.

## 3.1 PRELIMINARY

**Notation.** Let $\mathcal{X}$ denote an input feature space, with dimension $d$, and $\mathcal{Y}$ a target class label space. A domain, $\mathcal{D}$, is composed of data sampled from a joint distribution $\mathbb{P}(X, Y)$ on $\mathcal{X} \times \mathcal{Y}$, where $\mathcal{D} = (\boldsymbol{x}_i, y_i)_{i=1}^n \sim \mathbb{P}(X, Y)$, $\boldsymbol{x} \in \mathcal{X} \subset \mathbb{R}^d$, $y \in \mathcal{Y} \subset \mathbb{R}$ and $n$ is the number of data in the domain. Here, $X$ and $Y$ denote the corresponding random variables (Zhou et al., 2022a; Wang et al., 2022).

**Domain Generalization.** For the task of domain generalization, the input is $N$ source domains (training set), $\mathcal{S} = \{\mathcal{D}^j \mid j = 1, \cdots, N\}$, where $\mathcal{D}^j = \{(\boldsymbol{x}_i^j, y_i^j)\}_{i=1}^{n_j}$ denotes the $j^{th}$ domain and $n_j$ denotes the number of examples in $j^{th}$ domain. The joint distributions between each pair of domains are different: $\mathbb{P}(X, Y)^{(j)} \neq \mathbb{P}(X, Y)^{(k)}, j \neq k$. The goal of domain generalization is to learn a robust and generalizable predictive function $f : \mathcal{X} \to \mathcal{Y}$ from the $N$ source domains to achieve a minimum prediction error on an unseen target domain $\mathcal{T}$, where $\mathcal{T}$ cannot be accessed during training and $\mathbb{P}(X, Y)^{(\mathcal{T})} \neq \mathbb{P}(X, Y)^{(j)}$ for $j \in \{1, \cdots, N\}$.

**Knowledge Distillation.** Let $T(x)$ and $S(x)$ denote the outputs of the teacher and student models, respectively, for a given input $x$. The knowledge distillation loss $\mathcal{L}_{KD}$ is typically defined as the Kullback-Leibler (KL) divergence between the outputs of the teacher and student models: $\mathcal{L}_{KD} = \text{KL}\left(T(x) \parallel S(x)\right)$.

## 3.2 BALANCED ONLINE KNOWLEDGE DISTILLATION

**Teacher Model.** To enable the student to distil domain-specific knowledge, the teacher model must first acquire this knowledge. We employ Contrastive Language-Image Pretraining (CLIP) (Radford et al., 2021) as the backbone for the teacher model, including both an image encoder and a text encoder. CLIP was chosen because it demonstrated strong generalization capabilities in associating images with their corresponding textual descriptions. For extracting domain-invariant knowledge, the teacher model utilizes the pretrained image encoder without additional fine-tuning. To capture domain-specific knowledge, we incorporate adapters (Gao et al., 2024), a parameter-efficient tuning method, where each adapter is specialized for a specific domain. As illustrated in Figure 2, multiple domain expert adapters are appended to the image encoder, with the number of adapters corresponding to the number of source domains.

We use the cross-entropy loss, $\mathcal{L}_{\text{ce}}$, for each expert adapter $E$. Unlike relying solely on a similarity-based metric, cross-entropy loss inherently incorporates the calculation of similarity metrics (Radford et al., 2021). This approach enables us to both maximize the similarity between an image and its ground-truth prompt and minimize the similarity between the image and its unmatched class prompts, ensuring a more comprehensive optimization. For each class $c$, we generate $m \times N$ prompts in the format: "a picture of a $\{D^j\}\{c^k\}$.", where $D^j$ represents the $j$-th domain and $c^k$ represents the $k$-th class. The text encoder of the teacher model converts these prompts into text embeddings, yielding $m$ text embeddings per domain, corresponding to the $m$ classes. When processing an image from domain $D^i$, the corresponding expert adapter $E^i$ calculates the cross-entropy loss $\mathcal{L}_{\text{E}}$ for each domain, as defined in Equation 1. Here, $T_{\text{img}}$ denotes the image encoder of the teacher model, $E^j$ and $T^j$ represent the expert and text embeddings for the $j$-th domain, where $j \in \{1, \cdots, N\}$, and sim refers to the similarity measurement used to evaluate the similarity of image-text pairs. We adopt cosine similarity by following previous works (Radford et al., 2021).

$$\mathcal{L}_{\text{E}}^j = \mathcal{L}_{\text{ce}}(\text{sim}(E^j(T_{\text{img}}(x)), T^j), y) \tag{1}$$

After calculating $\mathcal{L}_{\text{E}}$ for each domain, BOLD computes the domain loss $\mathcal{L}_{\text{domain}}^i$ for expert adapter $i$ by minimizing the loss for its corresponding domain while maximizing the loss for other domains, as outlined in Equation 2.

$$\mathcal{L}_{\text{domain}}^i = \mathcal{L}_{\text{E}}^i - \frac{1}{N-1} \sum_{j=1, j \neq i}^{N} \mathcal{L}_{\text{E}}^j \tag{2}$$

**Student Model.** To distill both domain-invariant and domain-specific knowledge from the teacher model, we introduce two distillation losses: Invariant Distillation Loss ($\mathcal{L}_{\text{inv}}$) and Student-Specific Distillation Loss ($\mathcal{L}_{\text{sspc}}$), as defined in Equations 3 and 4. The loss $\mathcal{L}_{\text{inv}}$ minimizes the KL divergence between the outputs of the student model and the image encoder of the teacher model, while $\mathcal{L}_{\text{sspc}}^i$ minimizes the KL divergence between the outputs of the student model and the outputs of the relevant domain expert adapter $E^i$ corresponding to the domain of the input data. Since KL divergence is an asymmetric distance measure, the direction of distribution guidance is crucial. In our approach, the distribution of the teacher model's output is used to guide the output of the student model when distilling knowledge from the teacher model to the student model.

$$\mathcal{L}_{\text{inv}} = \text{KL}(T_{\text{img}}(x) \parallel S(x)) \tag{3}$$

$$\mathcal{L}_{\text{sspc}}^i = \text{KL}(E^i(T_{\text{img}}(x)) \parallel S(x)) \tag{4}$$

Additionally, the student model learns independently by minimizing the cross-entropy of the given input. The complete loss function is outlined in Equation 5.

$$\mathcal{L}_S = \mathcal{L}_{\text{inv}} + \mathcal{L}_{\text{sspc}} + \mathcal{L}_{\text{ce}}(S(x), y) \tag{5}$$

The combination of invariant and student-specific distillation losses enables the student model to capture both the common features shared across domains and the unique characteristics specific to each domain, which is crucial for enhancing the model's ability to generalize to unseen domains, particularly when the target domain shares characteristics with some of the source domains. Additionally, minimizing the divergence between the student and teacher outputs is a form of regularization, mitigating the risk of overfitting the source domain data. Since the teacher model's output represents a full probability distribution over all classes, the student learns not only to fit the correct label but also to approximate this probability distribution, which accounts for uncertainty. Furthermore, minimizing the divergence between the student and teacher outputs allows the student to capture implicit information encoded in the teacher's soft outputs about inter-class relationships. These relationships often include subtle correlations and patterns not apparent through hard labels (Wang et al., 2021).

**Online Distillation.** In contrast to existing knowledge distillation-based domain generalization methods that rely on a fixed teacher model, we adopt an online knowledge distillation strategy that allows the teacher model to adapt based on feedback from the student. To achieve this, we incorporate Teacher-Specific Distillation Loss ($\mathcal{L}_{\text{tspc}}$), defined in Equation 6 and incorporate it into the teacher model's learning objective, as shown in Equation 7. Unlike the Student-Specific Distillation Loss ($\mathcal{L}_{\textbf{sspc}}$), the Teacher-Specific Distillation Loss utilizes the output of the student model to guide the teacher model's output. During training, only the domain expert adapter corresponding to the domain of the input data is updated, while the image encoder of the teacher model remains

frozen. Domain expert adapters for domains not represented by the current input data are also un-affected. Here, $\mathcal{L}_T^i$ denotes the loss for the domain expert adapter associated with the $i$-th domain while $\mathcal{L}_{\text{domain}}^i$ and $\mathcal{L}_{\text{tspc}}^i$ are the domain and teacher-specific distillation loss for the $i$-th domain.

$$\mathcal{L}_{\text{tspc}}^i = \text{KL}(S(x) \parallel E^i(T_{\text{img}}(x))) \tag{6}$$

$$\mathcal{L}_T^i = \mathcal{L}_{\text{domain}}^i + \mathcal{L}_{\text{tspc}}^i \tag{7}$$

The online distillation strategy enables the teacher model to adapt in real-time based on feedback from the student model. Unlike fixed teacher models, which may become outdated as the student evolves, this dynamic adaptation ensures that the transferred knowledge remains relevant and continuously refined, resulting in more effective knowledge transfer. Moreover, the online distillation approach supports an end-to-end training process, eliminating the need for a separate training phase.

---

**Algorithm 1** Balanced Online Knowledge Distillation

1: **Input:** $\mathcal{D}$: training set; $T_{\text{img}}$: CLIP image encoder; $T_{\text{text}}$: CLIP text encoder; $N$: number of source domains.
2: **Output:** $S$: the optimal parameters for the student model.
3: Initialize $N$ expert adapters $\{E^1, \cdots, E^N\}$.
4: Generate $m \times N$ text embeddings $\{T_1^1, \cdots, T_m^1, T_1^2, \cdots T_m^N\}$ with $T_{\text{text}}$.
5: **for** each epoch **do**
6:    Random sampling $(x_i, y_i, d_i) \sim \mathcal{D}$.    ▷ Sample an input $x_i$ with class $y_i$ and domain $d_i$.
7:    $\mathcal{L}_E^j = \mathcal{L}_{\text{ce}}(\text{sim}(E^j(T_{\text{img}}(x_i)), T_{y_i}^j), y_i)$   ▷ Compute the cross-entropy loss for each domain.
8:    $\mathcal{L}_{\text{domain}}^{d_i} = \mathcal{L}_E^{d_i} - \dfrac{1}{N-1} \sum_{j=1, j \neq d_i}^N \mathcal{L}_E^j$    ▷ Compute domain loss for expert adapter $E^{d_i}$
9:    $\mathcal{L}_{\text{inv}} = \text{KL}(S(x_i) \parallel T_{\text{img}}(x_i))$    ▷ Compute the invariant distillation loss for $x_i$.
10:    $\mathcal{L}_{\text{sspc}}^{d_i} = \text{KL}(E^{d_i}(T_{\text{img}}(x_i)) \parallel S(x_i))$ ▷ Compute the student-specific distillation loss for $x_i$.
11:    $\mathcal{L}_{\text{tspc}}^{d_i} = \text{KL}(S(x_i) \parallel E^{d_i}(T_{\text{img}}(x_i)))$ ▷ Compute the teacher-specific distillation loss for $x_i$.
12:    $\mathcal{L}_S = \mathcal{L}_{\text{inv}} + \mathcal{L}_{\text{sspc}}^{d_i} + \mathcal{L}_{\text{ce}}(S(x_i), y_i)$    ▷ Combine invariant and specific distillation losses with the cross-entropy loss to calculate the overall loss for the student model.
13:    $\mathcal{L}_T^{d_i} = \mathcal{L}_{\text{domain}}^{d_i} + \mathcal{L}_{\text{tspc}}^{d_i}$    ▷ Combine domain loss with specific distillation loss (student feedback) to calculate the overall loss for domain expert adapter $E^{d_i}$ in the teacher model.
14:    Update the student model $S$ with $\mathcal{L}_S$.
15:    Update the domain expert adapter $E^{d_i}$ in the teacher model with $\mathcal{L}_T^{d_i}$.
16: **end for**
17: **Return** $S$

---

### 3.3 THEORETICAL DISCUSSION

This section explores the effectiveness of the proposed framework. As shown in Equation 8, the error bound for domain generalization can be decomposed into two key components: (1) the empirical risk within source domains and (2) the discrepancy between source and target domains. First, we demonstrate how incorporating domain-specific knowledge tightens the generalization error bound by reducing the discrepancy between source and target domains. Next, we show how online knowledge distillation further tightens the error bound by reducing the empirical risk in source domains.

**Effectiveness of Domain-Specific Knowledge for Domain Generalization**. In Domain Generalization, the error bound is commonly employed to evaluate a model's generalization performance on unseen domains. Within the Probably Approximately Correct (PAC)-Bayesian (McAllester, 1999) framework, the risk on the target domain $\mathcal{D}_T$ for any hypothesis $h \in \mathcal{H}$ can be bounded as follows:

$$L(h, \mathcal{D}_T) \leq \frac{1}{N} \sum_{i=1}^N L(h, \mathcal{D}_S^i) + \frac{1}{N} \sum_{i=1}^N d_{\mathcal{H}\Delta\mathcal{H}}(\mathcal{D}_S^i, \mathcal{D}_T) + \lambda, \tag{8}$$

where $\mathcal{H}$ denotes the hypothesis space containing all possible models, $\mathcal{D}_S$ represents the set of source domain distributions encompassing $N$ source domains, and $\mathcal{D}_T$ denotes the target domain

distribution. The term $L(h, \mathcal{D}_S^i)$ represents the risk of hypothesis $h$ on the $i$-th source domain, while $d_{\mathcal{H}\Delta\mathcal{H}}(\mathcal{D}_S^i, \mathcal{D}_T)$ indicates the discrepancy between the $i$-th source and target domains. Finally, $\lambda$ is a constant that reflects the model's complexity and its capacity for generalization.

Let $\mathcal{O}$ represent the function that quantifies the reduction in divergence caused by domain-specific knowledge shared between a source domain $D_S^i$ and the target domain $D_T$. We decompose the second term, $\sum_{i=1}^N d_{\mathcal{H}\Delta\mathcal{H}}(\mathcal{D}_S^i, \mathcal{D}_T)$, into two components, as shown in Equation 9. Here, $D_S^i \in S_{\text{no}}$ denotes the source domains that do not share domain-specific knowledge with the target domain (*i.e.*, $\mathcal{O}(D_S^i, D^T) = 0$), while $D_S^i \in S_{\text{o}}$ represents the source domains that share domain-specific knowledge with the target domain (*i.e.*, $\mathcal{O}(D_S^i, D^T) > 0$), and $N_{\text{no}} + N_{\text{o}} = N$.

$$\sum_{i=1}^N d_{\mathcal{H}\Delta\mathcal{H}}(\mathcal{D}_S^i, \mathcal{D}_T) = \sum_{i=1}^{N_{\text{no}}} d_{\mathcal{H}\Delta\mathcal{H}}(\mathcal{D}_S^i, \mathcal{D}_T) + \sum_{i=1}^{N_{\text{o}}} \left( d_{\mathcal{H}\Delta\mathcal{H}}(\mathcal{D}_S^i, \mathcal{D}_T) - \mathcal{O}(\mathcal{D}_S^i, \mathcal{D}_T) \right) \quad (9)$$

Since the term $\mathcal{O}(D_S^i, D^T) > 0$ for $D_S^i \in S_{\text{o}}$ is positive, its subtraction tightens the error bound, thereby indicating improved generalization performance. The decomposition of the divergence term provides two key insights regarding the role of domain-specific knowledge: (1) The greater the amount of domain-specific knowledge shared between the source and target domains, the tighter the error bound becomes. (2) As the number of source domains sharing domain-specific knowledge with the target domain increases, the error bound is further tightened.

**Effectiveness of Online Knowledge Distillation for Domain Generalization**. In this section, we first demonstrate how offline knowledge distillation reduces the empirical risk within source domains, followed by an explanation of how online knowledge distillation further minimizes this risk. Let $h_T$ denote the teacher model and $h$ the student model, which is the same model we aim to optimize in the previous analysis. The objective of the knowledge distillation loss is to minimize the discrepancy between the predictions of the student and teacher models across the source domains. This objective is formalized through the following loss minimization:

$$\min_h \frac{1}{N} \sum_{i=1}^N (L(h, D_S^i) + L_{\text{KD}}(h, h_T, D_S^i)), \quad (10)$$

where $L(h, D_S^i)$ represents the student's loss on the $i$-th source domain, and $L_{\text{KD}}(h, h_T, D_S^i)$ denotes the knowledge distillation loss between the student and teacher models on the $i$-th domain. Given that the teacher model is larger and pretrained on vast amounts of data, it typically outperforms the student model on the training data. Therefore, the student's loss on a source domain is expected to be higher than that of the teacher, leading to:

$$L(h, D_S^i) \geq L(h_T, D_S^i) + \epsilon, \quad (11)$$

where $\epsilon$ is a discrepancy reflecting the mismatch between the student and teacher. By substituting the student's loss on the source domains with this inequality, we derive a new error bound, as shown in Equation 12. Minimizing the knowledge distillation loss indirectly reduces the discrepancy $\epsilon$ between the student and teacher, resulting in a tighter error bound than the original.

$$L(h, \mathcal{D}_T) \leq \frac{1}{N} \sum_{i=1}^N (L(h_T, D_S^i) + \epsilon) + \frac{1}{N} \sum_{i=1}^N d_{\mathcal{H}\Delta\mathcal{H}}(\mathcal{D}_S^i, \mathcal{D}_T) + \lambda. \quad (12)$$

After demonstrating the effectiveness of offline knowledge distillation, we extend the analysis to online knowledge distillation. The key difference between offline and online distillation lies in the dynamic nature of the teacher model in the online setting, which evolves during training, providing more adaptive and real-time feedback to the student. This results in a modified inequality:

$$L(h, D_S^i) \geq L(h_T, D_S^i) + \epsilon_o. \quad (13)$$

Here, $\epsilon_o$ represents the average dynamic discrepancy between the student and teacher. Since the teacher model also minimizes the discrepancy between its output and the student's output, $\epsilon_o$ is expected to be smaller than $\epsilon$ in Equation 11, leading to $L(h_T, D_S^i) + \epsilon_o \leq L(h_T, D_S^i) + \epsilon$. Consequently, the error bound for domain generalization with online knowledge distillation is further tightened, as demonstrated in Equation 14.

$$L(h, \mathcal{D}_T) \leq \frac{1}{N} \sum_{i=1}^N (L(h_T, D_S^i) + \epsilon_o) + \frac{1}{N} \sum_{i=1}^N d_{\mathcal{H}\Delta\mathcal{H}}(\mathcal{D}_S^i, \mathcal{D}_T) + \lambda \quad (14)$$

In online knowledge distillation, where the teacher model is dynamically updated during training to minimize the discrepancy between its output and the student's, concerns may arise regarding the potential degradation of $L(h_T, D_S^i)$. However, because the teacher model typically initializes as a robust, pretrained model with strong generalization capabilities, these incremental updates are unlikely to degrade $L(h_T, D_S^i)$.

## 4 EXPERIMENTS

We evaluate our approach on seven domain generalization benchmarks, Digits (Zhou et al., 2020), PACS (Li et al., 2017), OfficeHome (Venkateswara et al., 2017), VLCS (Fang et al., 2013), Terra Incognita (Beery et al., 2018), NICO++ (Zhang et al., 2023a), and DomainNet (Peng et al., 2019), and compare it against several state-of-the-art domain generalization approaches, including Cross-Grad (Shankar et al., 2018), DDAIG (Zhou et al., 2020), MixStyle (Zhou et al., 2021), Domain-Mix (Sun et al., 2022), EFDMix (Zhang et al., 2022), RISE (Huang et al., 2023), SSPL (Zhao et al., 2024), and CLIP-Adapter (Gao et al., 2024) along with two additional baseline approaches: Empirical Risk Minimization (ERM) and Naive Knowledge Distillation (NKD) (Wang et al., 2021). Consistent with prior work (Zhou et al., 2022a), we adopt the leave-one-out evaluation strategy. Please refer to the reproducibility section in the appendix for details of experiment settings.

### 4.1 EXPERIMENTAL RESULTS

Table 1 shows leave-one-domain-out results for all benchmarks and baselines (excluding Cross-Grad). Full results are in the appendix. The best results are highlighted in bold.

**Overall accuracy across benchmarks.** Table 1 reveals three key findings: (1) BOLD consistently outperforms other state-of-the-art approaches, achieving the highest accuracy on the PACS, OfficeHome, VLCS, NICO++, and DomainNet datasets, demonstrating its effectiveness in enhancing model generalizability to unseen domains. Notably, BOLD's strong performance on the large-scale NICO++ and DomainNet datasets underscores its scalability. (2) It is important to note that knowledge distillation-based methods (NKD, RISE, and BOLD) exhibit weaker performance on the Terra Incognita and Digits datasets. This underperformance is attributed to the limitations of the teacher model, CLIP, which performs poorly on these datasets, resulting in weaker student performance as the students are trained to imitate the teacher's outputs. (3) Despite the overall lower performance on the Terra and Digits datasets, our method outperforms other knowledge distillation-based methods by a clear margin, with an improvement of approximately 10% on the Terra dataset. The strong performance of BOLD on Terra demonstrates the effectiveness of learning domain-specific knowledge in domain generalization, as the domain-specific characteristics in Terra, such as shape and colour, remain consistent across certain domains. This contrasts with datasets like PACS, where domain-specific characteristics are closely tied to visual styles that vary significantly across domains.

**Effectiveness across different backbones.** Table 2 presents the evaluation results of BOLD compared to NKD and RISE using different backbones on the PACS and OfficeHome datasets. For the complete results, please refer to the appendix. Table 2 includes results from distilling knowledge from ResNet50 and ViT-B/32 to ResNet18, as well as from ResNet50 to ResNet50. The evaluation of distilling knowledge from ViT-B/32 to ResNet50 is provided in Table 1. These results demonstrate that BOLD consistently outperforms other knowledge distillation-based domain generalization methods, regardless of the backbone, underscoring its effectiveness in domain generalization.

### 4.2 ABLATION STUDY

**Effectiveness of distilling domain-specific knowledge and online distillation strategy**. Table 3 presents the ablation study results, validating the effectiveness of distilling domain-specific knowledge and the online distillation strategy. Full results for all benchmarks are available in the appendix. Here, Invariant represents the results where the student model distils only domain-invariant knowledge from the teacher model. $Spc_{off}$ refers to the results where the student distils both domain-invariant and domain-specific knowledge from the teacher model but in an offline manner, where the teacher does not incorporate feedback from the student. $Spc_{on}$ represents the results where the student distils both domain-invariant and domain-specific knowledge in an online manner, where the teacher adapts to student feedback during training.

Table 1: Leave-one-domain-out accuracies on PACS, OfficeHome, VLCS, Terra Incognita, Digits, NICO++, and DomainNet. DomMix denotes DomainMix.

| | | ERM | DDAIG | MixStyle | DomMix | EDFMix | SSPL | NKD | RISE | BOLD |
|---|---|---|---|---|---|---|---|---|---|---|
| **PACS** | Art | 81.0 ± .3 | 83.4 ± .4 | 84.2 ± .2 | 85.9 ± .4 | 87.2 ± .3 | **87.9 ± .2** | 82.5 ± .2 | 85.7 ± .2 | **88.1 ± .2** |
| | Cartoon | 74.0 ± .5 | 74.7 ± .3 | 74.5 ± .3 | 72.8 ± .5 | 76.1 ± .5 | 76.9 ± .3 | 83.3 ± .5 | 85.2 ± .4 | **86.9 ± .3** |
| | Photo | 96.2 ± .3 | 96.8 ± .3 | **97.8 ± .2** | 97.1 ± .2 | **98.0 ± .1** | 97.8 ± .2 | 97.2 ± .2 | 97.4 ± .4 | 97.9 ± .2 |
| | Sketch | 71.0 ± .5 | 73.7 ± .3 | 72.6 ± .4 | 73.6 ± .3 | 76.9 ± .3 | 77.5 ± .3 | 75.6 ± .2 | 78.2 ± .3 | **78.8 ± .3** |
| | Avg. | 80.8 ± .4 | 82.1 ± .3 | 82.3 ± .3 | 82.3 ± .3 | 84.6 ± .4 | 85.0 ± .2 | 84.6 ± .3 | 86.6 ± .3 | **87.9 ± .3** |
| **OfficeHome** | Artistic | 67.1 ± .2 | 68.8 ± .3 | 68.6 ± .3 | 69.0 ± .2 | 69.1 ± .2 | 69.4 ± .1 | 68.7 ± .2 | **69.5 ± .1** | **69.7 ± .2** |
| | Clipart | 55.1 ± .2 | 53.4 ± .2 | 55.4 ± .3 | 54.6 ± .4 | 57.1 ± .1 | 58.3 ± .2 | 54.7 ± .2 | 55.8 ± .2 | **58.9 ± .3** |
| | Product | 78.2 ± .3 | 78.0 ± .2 | 78.9 ± .2 | 77.5 ± .2 | 79.1 ± .1 | **79.7 ± .2** | 79.5 ± .3 | **79.7 ± .3** | 80.1 ± .2 |
| | RealWorld | 82.0 ± .1 | 81.2 ± .1 | 82.3 ± .1 | 81.5 ± .2 | 82.3 ± .2 | 81.6 ± .3 | 82.3 ± .1 | 82.6 ± .3 | **83.3 ± .2** |
| | Avg. | 70.6 ± .2 | 70.3 ± .2 | 71.3 ± .2 | 70.7 ± .3 | 71.9 ± .2 | 72.3 ± .2 | 71.3 ± .2 | 71.9 ± .2 | **73.0 ± .2** |
| **VLCS** | Caltech | 97.2 ± .2 | 97.8 ± .3 | 98.0 ± .2 | 97.2 ± .2 | 98.0 ± .4 | 98.2 ± .3 | **99.4 ± .3** | **99.5 ± .3** | 99.5 ± .2 |
| | LabelMe | 61.1 ± .1 | 61.9 ± .4 | 60.5 ± .3 | 59.2 ± .2 | 61.3 ± .2 | 62.1 ± .2 | 65.7 ± .3 | 66.1 ± .4 | **68.2 ± .3** |
| | Pascal | 77.0 ± .2 | 75.8 ± .3 | 76.6 ± .3 | 69.8 ± .3 | 77.1 ± .3 | 77.7 ± .2 | 82.3 ± .2 | 82.7 ± .3 | **83.5 ± .3** |
| | Sun | 66.8 ± .2 | 67.6 ± .3 | 67.4 ± .4 | 70.4 ± .3 | 67.6 ± .3 | 69.9 ± .3 | 70.6 ± .1 | 71.3 ± .2 | **71.7 ± .2** |
| | Avg. | 75.5 ± .2 | 75.8 ± .3 | 75.6 ± .3 | 74.2 ± .3 | 76.0 ± .3 | 77.0 ± .3 | 79.5 ± .2 | 79.9 ± .3 | **80.7 ± .2** |
| **Terra** | L38 | 20.4 ± .1 | 25.9 ± .3 | 36.2 ± .5 | 33.0 ± .4 | **38.0 ± .4** | **38.1 ± .3** | 25.5 ± .4 | 26.5 ± .3 | **38.3 ± .3** |
| | L43 | 35.2 ± .2 | 33.8 ± .4 | 43.5 ± .3 | 38.1 ± .3 | **44.6 ± .4** | **44.6 ± .3** | 29.3 ± .3 | 30.6 ± .2 | 41.2 ± .2 |
| | L46 | 29.6 ± .2 | 28.9 ± .3 | 32.2 ± .3 | **34.5 ± .2** | 31.9 ± .4 | **34.6 ± .3** | 25.5 ± .2 | 27.0 ± .4 | 31.5 ± .2 |
| | L100 | 39.0 ± .3 | 40.6 ± .3 | **54.9 ± .4** | 45.9 ± .3 | 52.3 ± .1 | **55.0 ± .4** | 18.0 ± .3 | 19.4 ± .4 | 28.1 ± .2 |
| | Avg. | 31.1 ± .2 | 32.3 ± .3 | 41.7 ± .4 | 37.9 ± .3 | 41.7 ± .3 | **43.1 ± .3** | 24.6 ± .3 | 25.9 ± .3 | 34.8 ± .2 |
| **Digits** | MNIST | 96.5 ± .1 | 96.7 ± .3 | **97.5 ± .3** | 96.7 ± .4 | **97.6 ± .2** | 97.6 ± .2 | 71.6 ± .2 | 72.1 ± .2 | 74.3 ± .3 |
| | MNIST-M | 64.2 ± .4 | 66.1 ± .5 | 67.3 ± .4 | 67.0 ± .4 | **68.1 ± .2** | **68.2 ± .1** | 40.8 ± .3 | 41.4 ± .3 | 43.2 ± .3 |
| | SVHN | 70.3 ± .3 | 70.5 ± .4 | **70.8 ± .2** | 69.2 ± .4 | **70.7 ± .1** | **70.8 ± .1** | 30.3 ± .4 | 31.3 ± .3 | 34.3 ± .2 |
| | SYN | 88.2 ± .3 | 89.8 ± .3 | 90.3 ± .2 | 86.6 ± .4 | 90.3 ± .2 | **90.6 ± .2** | 58.7 ± .4 | 62.3 ± .2 | 64.5 ± .3 |
| | Avg. | 79.8 ± .3 | 80.8 ± .4 | 81.5 ± .2 | 79.9 ± .4 | **81.7 ± .2** | **81.8 ± .1** | 50.4 ± .3 | 51.8 ± .2 | 54.1 ± .3 |
| **NICO++** | Autumn | 82.9 ± .1 | 84.8 ± .4 | 85.1 ± .1 | 86.3 ± .2 | 85.6 ± .2 | 86.0 ± .2 | 85.0 ± .3 | 85.9 ± .3 | **86.9 ± .4** |
| | Dim | 75.8 ± .1 | 78.3 ± .2 | 79.3 ± .2 | 81.1 ± .2 | 80.4 ± .1 | 80.9 ± .2 | 78.5 ± .4 | 80.9 ± .3 | **81.6 ± .2** |
| | Grass | 84.9 ± .3 | 86.5 ± .2 | 87.0 ± .3 | 87.8 ± .3 | 87.5 ± .3 | 87.7 ± .1 | 85.8 ± .3 | 87.2 ± .3 | **88.3 ± .3** |
| | Outdoor | 82.4 ± .1 | 83.3 ± .2 | 84.8 ± .3 | 85.3 ± .1 | 85.3 ± .2 | 85.1 ± .2 | 83.4 ± .4 | 84.9 ± .2 | **85.7 ± .4** |
| | Rock | 83.9 ± .4 | 85.3 ± .1 | 85.9 ± .1 | 87.0 ± .1 | 86.4 ± .3 | 87.0 ± .2 | 84.4 ± .4 | 85.8 ± .3 | **87.4 ± .4** |
| | Water | 77.5 ± .2 | 78.8 ± .2 | 79.7 ± .1 | 80.3 ± .2 | 80.3 ± .1 | 80.1 ± .2 | 78.8 ± .4 | 80.2 ± .3 | **81.9 ± .4** |
| | Avg. | 81.2 ± .2 | 82.8 ± .2 | 83.7 ± .2 | 84.6 ± .2 | 84.3 ± .2 | 84.5 ± .2 | 82.6 ± .4 | 84.2 ± .3 | **85.3 ± .3** |
| **DomainNet** | Clipart | 63.4 ± .2 | 61.3 ± .2 | 63.9 ± .2 | 63.5 ± .2 | 64.2 ± .2 | 63.9 ± .2 | 63.9 ± .2 | 64.3 ± .1 | **64.8 ± .2** |
| | Infograph | 25.8 ± .3 | 27.9 ± .2 | 29.7 ± .1 | 27.5 ± .3 | 30.8 ± .2 | 31.0 ± .2 | 34.9 ± .2 | 35.0 ± .2 | **36.7 ± .2** |
| | Painting | 49.7 ± .2 | 51.4 ± .2 | 54.2 ± .1 | 53.1 ± .1 | 54.6 ± .2 | 55.2 ± .2 | 56.3 ± .2 | 57.2 ± .1 | **60.2 ± .2** |
| | Quickdraw | 11.8 ± .3 | 10.1 ± .1 | 11.7 ± .2 | 10.9 ± .1 | **12.3 ± .2** | 12.9 ± .2 | 10.1 ± .3 | 10.8 ± .2 | **12.3 ± .3** |
| | Real | 61.6 ± .3 | 61.0 ± .2 | 64.1 ± .2 | 63.4 ± .2 | 64.5 ± .2 | 64.3 ± .1 | 71.9 ± .3 | 72.6 ± .2 | **75.4 ± .3** |
| | Sketch | 48.1 ± .2 | 50.6 ± .2 | 52.9 ± .2 | 52.1 ± .2 | 53.6 ± .2 | 53.2 ± .2 | 50.5 ± .2 | 52.4 ± .3 | **55.9 ± .4** |
| | Avg. | 43.4 ± .2 | 43.7 ± .2 | 46.1 ± .2 | 45.1 ± .2 | 46.7 ± .2 | 46.8 ± .2 | 47.9 ± .2 | 48.7 ± .2 | **50.9 ± .2** |

Based on the results in Table 3, we make three key observations: (1) When the invariant knowledge across source and target domain is highly representative, such as the Photo domain of PACS, the RealWorld domain of OfficeHome, and the Caltech domain of VLCS, the improvements from distilling domain-specific knowledge and applying the online knowledge distillation strategy are relatively minor. (2) When the target domain shares specific characteristics with the source domains, the improvement from distilling domain-specific knowledge is substantial, as evidenced by the Terra dataset discussed in the previous section. (3) The effectiveness of the online knowledge distillation strategy is closely correlated with the capability of the teacher model. For instance, in the Terra dataset, the improvement of online knowledge distillation is minor due to the teacher's limited performance. Conversely, in the Clipart domain of OfficeHome, while the improvement from domain-specific knowledge is limited, the presence of a strong teacher model results in significant gains from the online knowledge distillation strategy. For further details on the teacher model's performance across benchmarks, please refer to the CLIP zero-shot results provided in the appendix.

**T-SNE Visualization.** Figure 3 presents the T-SNE visualization for ERM, NKD, RISE, and BOLD on the PACS dataset. As shown, distilling knowledge from a large teacher model allows NKD, RISE, and BOLD to produce a more separable embedding space than ERM, highlighting the effectiveness of knowledge distillation. Furthermore, by incorporating domain-specific knowledge,

Table 2: Leave-one-domain-out accuracies on PACS and OfficeHome datasets for various knowledge distillation-based domain generalization approaches using different backbones.

| | | ResNet50 → ResNet18 | | | ViT-B/32 → ResNet18 | | | ResNet50 → ResNet50 | | |
| | | NKD | RISE | BOLD | NKD | RISE | BOLD | NKD | RISE | BOLD |
|---|---|---|---|---|---|---|---|---|---|---|
| **PACS** | Art | 77.4 ± .3 | 79.0 ± .2 | **80.2 ± .2** | 78.1 ± .4 | 79.3 ± .3 | **81.2 ± .2** | 80.7 ± .2 | 83.2 ± .3 | **85.1 ± .2** |
| | Cartoon | 76.4 ± .2 | 77.1 ± .2 | **77.6 ± .2** | 79.6 ± .3 | 81.1 ± .2 | **81.8 ± .2** | 80.9 ± .5 | 82.1 ± .4 | **82.6 ± .2** |
| | Photo | 94.1 ± .3 | 94.9 ± .3 | **95.3 ± .2** | 94.2 ± .2 | 95.6 ± .2 | **95.9 ± .2** | 95.2 ± .2 | 96.7 ± .2 | **97.0 ± .2** |
| | Sketch | 71.1 ± .2 | 72.9 ± .3 | **74.7 ± .2** | 72.8 ± .3 | 73.2 ± .3 | **76.8 ± .3** | 76.5 ± .5 | 78.0 ± .4 | **78.1 ± .2** |
| | Avg. | 79.7 ± .3 | 80.9 ± .2 | **82.0 ± .2** | 81.2 ± .3 | 82.3 ± .2 | **83.9 ± .2** | 83.3 ± .4 | 85.0 ± .3 | **85.7 ± .2** |
| **OfficeHome** | Artistic | 57.5 ± .4 | 58.0 ± .2 | **60.0 ± .2** | 58.4 ± .3 | 59.3 ± .1 | **60.6 ± .2** | 68.8 ± .3 | 69.3 ± .2 | **69.7 ± .2** |
| | Clipart | 48.0 ± .2 | 49.2 ± .1 | **54.9 ± .2** | 49.1 ± .2 | 51.0 ± .3 | **56.2 ± .2** | 55.3 ± .4 | 55.6 ± .3 | **58.5 ± .2** |
| | Product | 72.5 ± .3 | 73.2 ± .2 | **73.8 ± .1** | 72.7 ± .2 | 73.4 ± .3 | **73.9 ± .3** | 78.4 ± .1 | 78.9 ± .2 | **79.7 ± .2** |
| | RealWorld | 75.7 ± .1 | 75.8 ± .2 | **76.0 ± .1** | 76.0 ± .2 | 76.3 ± .2 | **77.0 ± .2** | 82.0 ± .3 | 82.2 ± .1 | **82.6 ± .1** |
| | Avg. | 63.4 ± .2 | 64.1 ± .2 | **66.2 ± .2** | 64.0 ± .2 | 65.0 ± .2 | **66.9 ± .2** | 71.1 ± .3 | 71.5 ± .2 | **72.6 ± .2** |

Table 3: Ablation study results on the PACS, OfficeHome, VLCS, and Terra Incognita datasets.

| | | Invariant | Spc$_{off}$ | Spc$_{on}$ | | | Invariant | Spc$_{off}$ | Spc$_{on}$ |
|---|---|---|---|---|---|---|---|---|---|
| **PACS** | Art | 82.5 ± .2 | 86.9 ± .3 | 88.1 ± .2 | **VLCS** | Caltech | 99.4 ± .3 | 99.5 ± .1 | 99.5 ± .2 |
| | Cartoon | 83.3 ± .5 | 85.7 ± .2 | 86.9 ± .3 | | LabelMe | 65.7 ± .3 | 66.4 ± .3 | 68.2 ± .3 |
| | Photo | 97.2 ± .2 | 97.3 ± .2 | 97.9 ± .2 | | Pascal | 82.3 ± .2 | 83.1 ± .2 | 83.5 ± .3 |
| | Sketch | 75.6 ± .2 | 77.1 ± .2 | 78.8 ± .3 | | Sun | 70.6 ± .1 | 71.4 ± .2 | 71.7 ± .2 |
| | Avg. | 84.6 ± .3 | 86.8 ± .2 | 87.9 ± .3 | | Avg. | 79.5 ± .2 | 80.1 ± .2 | 80.7 ± .2 |
| **OfficeHome** | Artistic | 68.7 ± .2 | 69.5 ± .1 | 69.7 ± .2 | **Terra** | L38 | 25.5 ± .4 | 35.3 ± .3 | 38.3 ± .3 |
| | Clipart | 54.7 ± .2 | 55.1 ± .2 | 58.9 ± .3 | | L43 | 29.3 ± .2 | 38.9 ± .3 | 41.2 ± .2 |
| | Product | 79.5 ± .3 | 79.9 ± .3 | 80.1 ± .2 | | L46 | 25.5 ± .2 | 30.1 ± .2 | 31.5 ± .2 |
| | RealWorld | 82.3 ± .1 | 82.6 ± .2 | 83.3 ± .2 | | L100 | 18.0 ± .3 | 26.2 ± .3 | 28.1 ± .2 |
| | Avg. | 71.3 ± .2 | 71.8 ± .2 | 73.0 ± .2 | | Avg. | 24.6 ± .3 | 32.6 ± .3 | 34.8 ± .2 |

BOLD achieves an even more distinct and well-separated embedding space than NKD and RISE, demonstrating the potential of domain-specific knowledge for effective domain generalization.

**Scalability.** We compared the parameter count across different backbones and expert adapters relative to the number of source domains. While the parameter count for the expert adapters increases linearly with the number of source domains, it remains negligible compared to the large parameter count in backbones, demonstrating BOLD's scalability. See the appendix for detailed results.

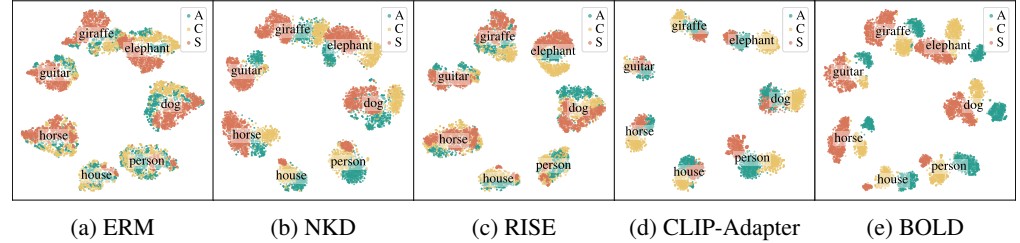

(a) ERM            (b) NKD            (c) RISE            (d) CLIP-Adapter            (e) BOLD

Figure 3: T-SNE visualization. A, C, and S denote Art, Cartoon, and Sketch domains, respectively.

## 5 CONCLUSION

Our Balanced Online Knowledge Distillation framework (BOLD)leverages domain-invariant and domain-specific knowledge through an online distillation strategy to enhance domain generalization. Theoretical analysis highlights the benefits of domain-specific knowledge when the target domain shares characteristics with source domains, as well as the advantages of the online distillation strategy for domain generalization. Extensive experiments across seven benchmarks and ablation studies validate the effectiveness and the effectiveness of the proposed framework. Future work will focus on developing more effective distillation strategies to overcome the teacher model's limited capability and expanding BOLD's application to more complex tasks like objective detection.

## 6 REPRODUCIBILITY STATEMENT

All details necessary to reproduce our experiments, including descriptions of the datasets, evaluation metrics, baselines, and training settings, are provided in the Appendix A.1. We have also included a complete description of the proposed BOLD framework and its training procedure in Section 3.2, with additional algorithmic details in Algorithm 1. The theoretical analysis, along with explanations of any assumptions, are available in Section 3.3. Furthermore, to facilitate reproducibility, we provide an anonymized link to the code in the abstract, which is also uploaded to the supplementary.

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

# A APPENDIX

## A.1 REPRODUCIBILITY

**Datasets.** The proposed approach is evaluated on seven domain generalization benchmarks, covering a range of image classification tasks: (1) **Digits** (Zhou et al., 2020) includes four digit recognition tasks: MNIST, MNIST-M, SVHN, and SYN. (2) **PACS** (Li et al., 2017) comprises four domains: Photo, Art Painting, Cartoon, and Sketch. (3) **OfficeHome** (Venkateswara et al., 2017) contains four domains: Artistic, Clipart, Product, and Real World, with 65 classes related to office and home objects. (4) **VLCS** (Fang et al., 2013) is collected from four domains: Caltech101, PASCAL, LabelMe, and Sun, featuring five common categories: bird, car, chair, dog, and person. (5) **Terra Incognita** (Beery et al., 2018) is a subset of the Caltech Camera Traps dataset, consists of four domains (L38, L43, L46, L100) representing different geographic locations, and includes nine species of wild animals along with a 'no-animal' class. (6) **NICO++** (Zhang et al., 2023a) is a recently constructed dataset from 2023 for out-of-distribution (OOD) image classification, comprising six domains and a total of 88,866 images. (7) **DomainNet** (Peng et al., 2019) is the largest domain generalization dataset, consisting of six domains, 345 classes, and 596,010 images.

**Baselines.** We evaluate our method against several state-of-the-art domain generalization approaches, including CrossGrad (Shankar et al., 2018), DDAIG (Zhou et al., 2020), MixStyle (Zhou et al., 2021), DomainMix (Sun et al., 2022), EFDMix (Zhang et al., 2022), RISE (Huang et al., 2023), and SSPL (Zhao et al., 2024). In addition, we evaluate two baseline methods: Empirical Risk Minimization (ERM), which aggregates data from all source domains without employing specialized domain generalization techniques, and Naive Knowledge Distillation (NKD) (Wang et al., 2021), which distils knowledge from the teacher model using only invariant distillation loss.

**Evaluation Metrics.** Following previous work (Zhou et al., 2022a), we adopt the leave-one-out evaluation strategy. Specifically, one domain is selected as the test domain, while the remaining domains are used as source domains for training. Performance is reported as the top-1 classification accuracy (%) averaged over ten runs, along with the corresponding 95% confidence intervals.

**Network Structure & Training.** For all benchmarks, input images are resized to $224 \times 224$ pixels, and the pretrained ResNet50 model is used as the backbone. ResNet50 also serves as the backbone for the student model in knowledge-distillation-based methods. For the teacher model, we evaluate both ResNet50 and ViT-B/32 from CLIP's image encoder to demonstrate the effectiveness of the proposed method. The implementation is carried out using the PyTorch library. Stochastic Gradient Descent (SGD) is used as the optimizer for training the baseline, student, and teacher models with a momentum of 0.9 and a weight decay of $5 \times 10^{-4}$. Across all benchmarks, the models are trained with a learning rate of 0.01 using the CosineAnnealingLR learning scheduler and a batch size with 64 for 50 epochs. All experiments are conducted on NVIDIA Tesla A100 80 GB GPUs.

## A.2 EXPERIMENTAL RESULTS

**Scalability.** Figure 4 presents the comparison of the parameter count across different backbones and expert adapters relative to the number of source domains. While the parameter count for the expert adapters increases linearly with the number of source domains, it remains negligible compared to the large parameter count in backbones, demonstrating BOLD's scalability.

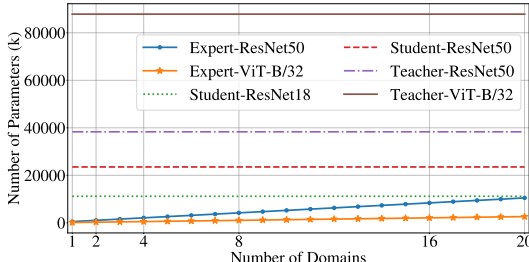

Figure 4: Scalability. Comparing the number of parameters across different backbones.

**Teacher Capability.** Table 4 presents the CLIP-ZeroShot accuracy across all benchmarks using ResNet50 and ViT-B/32 backbones, highlighting CLIP's performance across different benchmarks. As shown, CLIP, pretrained on 400 million images through Language-Image pertaining, demonstrates remarkable performance on domains such as Art and Photo in PACS and Caltech in VLCS. However, its performance significantly reduces on more abstract datasets like Terra and Digits.

Table 4: CLIP-ZeroShot accuracy across all benchmarks using ResNet50 and ViT-B/32 backbones.

| | | RN50 | ViT | | | RN50 | ViT | | | RN50 | ViT | | | RN50 | ViT |
|---|---|---|---|---|---|---|---|---|---|---|---|---|---|---|---|
| **PACS** | Art | 92.72 | 96.53 | **OfficeHome** | Artistic | 69.43 | 75.20 | **VLCS** | Caltech | 100.00 | 100.00 | **Terra** | L38 | 26.88 | 20.06 |
| | Cartoon | 94.88 | 98.12 | | Clipart | 54.20 | 66.41 | | LabelMe | 65.62 | 70.39 | | L43 | 32.64 | 30.08 |
| | Photo | 99.46 | 99.82 | | Product | 80.69 | 86.21 | | Pascal | 85.09 | 85.09 | | L46 | 24.38 | 19.14 |
| | Sketch | 80.07 | 85.46 | | RealWorld | 81.68 | 86.46 | | Sun | 71.47 | 72.18 | | L100 | 12.47 | 15.21 |
| | Avg. | 91.78 | 94.98 | | Avg. | 71.50 | 78.57 | | Avg. | 80.55 | 81.92 | | Avg. | 24.09 | 21.12 |

| | | RN50 | ViT | | | RN50 | ViT | | | RN50 | ViT |
|---|---|---|---|---|---|---|---|---|---|---|---|
| **NICO++** | Autumn | 83.16 | 86.11 | **DomainNet** | Clipart | 55.05 | 68.17 | **Digits** | MNIST | 35.40 | 25.12 |
| | Dim | 78.28 | 83.77 | | Infograph | 41.45 | 42.32 | | MNIST-M | 21.05 | 15.15 |
| | Grass | 86.13 | 87.92 | | Painting | 54.74 | 63.01 | | SVHN | 19.92 | 13.35 |
| | Outdoor | 81.00 | 82.75 | | Quickdraw | 6.13 | 13.09 | | SYN | 43.78 | 20.37 |
| | Rock | 83.46 | 85.12 | | Real | 77.94 | 81.33 | | Avg. | 30.04 | 18.50 |
| | Water | 72.80 | 75.96 | | Sketch | 49.65 | 58.57 | | | | |
| | Avg. | 80.81 | 83.61 | | Avg. | 47.49 | 54.42 | | | | |

Table 5 presents the ablation study results on the NICO++, DomainNet, and Digits datasets, which are not included in the main paper. Table 6 provides the complete evaluation results across all benchmarks and all baselines, including CrossGrad. Table 7 displays the evaluation results on the VLCS, Terra Incognita, NICO++, DomainNet, and Digits datasets for various knowledge distillation-based domain generalization approaches using different backbones.

Table 5: Ablation study results on the NICO++, DomainNet, and Digits datasets.

| | | Invariant | Spc$_{off}$ | Spc$_{on}$ | | | Invariant | Spc$_{off}$ | Spc$_{on}$ |
|---|---|---|---|---|---|---|---|---|---|
| **NICO++** | Autumn | 85.0 ± .3 | 86.3 ± .2 | 86.9 ± .4 | **DomainNet** | Clipart | 63.9 ± .2 | 64.6 ± .2 | 64.8 ± .2 |
| | Dim | 78.5 ± .4 | 80.4 ± .2 | 81.6 ± .2 | | Infograph | 34.9 ± .2 | 36.1 ± .3 | 36.7 ± .2 |
| | Grass | 85.8 ± .3 | 87.1 ± .3 | 88.3 ± .3 | | Painting | 56.3 ± .2 | 59.4 ± .3 | 60.2 ± .2 |
| | Outdoor | 83.4 ± .4 | 85.2 ± .2 | 85.7 ± .4 | | Quickdraw | 10.1 ± .3 | 11.9 ± .3 | 12.3 ± .3 |
| | Rock | 84.4 ± .4 | 86.3 ± .2 | 87.4 ± .4 | | Real | 71.9 ± .3 | 74.5 ± .2 | 75.4 ± .3 |
| | Water | 78.8 ± .4 | 80.0 ± .3 | 81.9 ± .4 | | Sketch | 50.5 ± .2 | 53.7 ± .4 | 55.9 ± .4 |
| | Avg. | 82.6 ± .4 | 84.2 ± .2 | 85.3 ± .3 | | Avg. | 47.9 ± .2 | 50.0 ± .3 | 50.9 ± .3 |
| **Digits** | MNIST | 71.6 ± .2 | 73.9 ± .3 | 74.3 ± .3 | | | | | |
| | MNIST-M | 40.8 ± .3 | 43.0 ± .2 | 43.2 ± .3 | | | | | |
| | SVHN | 30.3 ± .4 | 33.6 ± .2 | 34.3 ± .2 | | | | | |
| | SYN | 58.7 ± .4 | 62.8 ± .3 | 64.5 ± .3 | | | | | |
| | Avg. | 50.4 ± .3 | 53.3 ± .3 | 54.1 ± .3 | | | | | |

Table 6: Leave-one-domain-out accuracies on PACS, OfficeHome, VLCS, Terra Incognita, Digits, NICO++, and DomainNet. CroGrad and DomMix denote CrossGrad and DomainMix.

| | | ERM | CroGrad | DDAIG | MixStyle | DomMix | EDFMix | SSPL | NKD | RISE | BOLD |
|---|---|---|---|---|---|---|---|---|---|---|---|
| **PACS** | Art | 81.0 ± .3 | 81.9 ± .3 | 83.4 ± .4 | 84.2 ± .2 | 85.9 ± .4 | 87.2 ± .3 | **87.9 ± .2** | 82.5 ± .2 | 85.7 ± .2 | **88.1 ± .2** |
| | Cartoon | 74.0 ± .5 | 72.9 ± .2 | 74.7 ± .3 | 74.5 ± .3 | 72.8 ± .5 | 76.1 ± .5 | 76.9 ± .3 | 83.3 ± .5 | 85.2 ± .4 | **86.9 ± .3** |
| | Photo | 96.2 ± .3 | 96.6 ± .2 | 96.8 ± .3 | **97.8 ± .2** | 97.1 ± .2 | **98.0 ± .1** | 97.8 ± .2 | 97.2 ± .2 | 97.4 ± .4 | **97.9 ± .2** |
| | Sketch | 71.0 ± .5 | 71.5 ± .3 | 73.7 ± .3 | 72.6 ± .4 | 73.6 ± .3 | 76.9 ± .3 | 77.5 ± .3 | 75.6 ± .2 | 78.2 ± .3 | **78.8 ± .3** |
| | Avg. | 80.8 ± .4 | 80.8 ± .3 | 82.1 ± .3 | 82.3 ± .3 | 82.3 ± .3 | 84.6 ± .4 | 85.0 ± .2 | 84.6 ± .3 | 86.6 ± .3 | **87.9 ± .3** |
| **OfficeHome** | Artistic | 67.1 ± .2 | 67.5 ± .2 | 68.8 ± .3 | 68.6 ± .3 | 69.0 ± .2 | 69.1 ± .2 | 69.4 ± .1 | 68.7 ± .2 | **69.5 ± .1** | **69.7 ± .2** |
| | Clipart | 55.1 ± .2 | 54.9 ± .2 | 53.4 ± .2 | 55.4 ± .3 | 54.6 ± .4 | 57.1 ± .1 | 58.3 ± .2 | 54.7 ± .2 | 55.8 ± .2 | **58.9 ± .3** |
| | Product | 78.2 ± .3 | 78.2 ± .2 | 78.0 ± .2 | 78.9 ± .2 | 77.5 ± .2 | 79.1 ± .1 | **79.7 ± .2** | 79.5 ± .3 | **79.7 ± .3** | **80.1 ± .2** |
| | RealWorld | 82.0 ± .1 | 81.7 ± .2 | 81.2 ± .1 | 82.3 ± .1 | 81.5 ± .2 | 82.3 ± .2 | 81.6 ± .3 | 82.3 ± .1 | 82.6 ± .3 | **83.3 ± .2** |
| | Avg. | 70.6 ± .2 | 70.6 ± .2 | 70.3 ± .2 | 71.3 ± .2 | 70.7 ± .3 | 71.9 ± .2 | 72.3 ± .2 | 71.3 ± .2 | 71.9 ± .2 | **73.0 ± .2** |
| **VLCS** | Caltech | 97.2 ± .2 | 97.6 ± .4 | 97.8 ± .3 | 98.0 ± .2 | 97.2 ± .2 | 98.0 ± .4 | 98.2 ± .3 | **99.4 ± .3** | **99.5 ± .3** | **99.5 ± .2** |
| | LabelMe | 61.1 ± .1 | 60.9 ± .3 | 61.9 ± .4 | 60.5 ± .3 | 59.2 ± .2 | 61.3 ± .2 | 62.1 ± .2 | 65.7 ± .3 | 66.1 ± .4 | **68.2 ± .3** |
| | Pascal | 77.0 ± .2 | 76.7 ± .3 | 75.8 ± .3 | 76.6 ± .3 | 69.8 ± .3 | 77.1 ± .3 | 77.7 ± .2 | 82.3 ± .2 | 82.7 ± .3 | **83.5 ± .3** |
| | Sun | 66.8 ± .2 | 66.8 ± .2 | 67.6 ± .3 | 67.4 ± .4 | 70.4 ± .3 | 67.6 ± .3 | 69.9 ± .3 | 70.6 ± .1 | 71.3 ± .2 | **71.7 ± .2** |
| | Avg. | 75.5 ± .2 | 75.5 ± .3 | 75.8 ± .3 | 75.6 ± .3 | 74.2 ± .3 | 76.0 ± .3 | 77.0 ± .3 | 79.5 ± .2 | 79.9 ± .3 | **80.7 ± .2** |
| **Terra** | L38 | 20.4 ± .1 | 25.5 ± .2 | 25.9 ± .3 | 36.2 ± .5 | 33.0 ± .4 | **38.0 ± .4** | **38.1 ± .3** | 25.5 ± .4 | 26.5 ± .3 | **38.3 ± .3** |
| | L43 | 35.2 ± .2 | 35.1 ± .2 | 33.8 ± .4 | 43.5 ± .3 | 38.1 ± .2 | **44.6 ± .4** | **44.6 ± .3** | 29.3 ± .3 | 30.6 ± .2 | 41.2 ± .2 |
| | L46 | 29.6 ± .2 | 28.4 ± .4 | 28.9 ± .3 | 32.2 ± .3 | **34.5 ± .2** | 31.9 ± .4 | **34.6 ± .3** | 25.5 ± .2 | 27.0 ± .4 | 31.5 ± .2 |
| | L100 | 39.0 ± .3 | 39.4 ± .2 | 40.6 ± .3 | **54.9 ± .4** | 45.9 ± .3 | 52.3 ± .1 | **55.0 ± .4** | 18.0 ± .3 | 19.4 ± .4 | 28.1 ± .2 |
| | Avg. | 31.1 ± .2 | 32.1 ± .2 | 32.3 ± .3 | 41.7 ± .4 | 37.9 ± .3 | 41.7 ± .3 | **43.1 ± .3** | 24.6 ± .3 | 25.9 ± .3 | 34.8 ± .2 |
| **Digits** | MNIST | 96.5 ± .1 | 96.5 ± .3 | 96.7 ± .3 | **97.5 ± .3** | 96.7 ± .4 | **97.6 ± .2** | **97.6 ± .2** | 71.6 ± .2 | 72.1 ± .2 | 74.3 ± .3 |
| | MNIST-M | 64.2 ± .4 | 64.5 ± .3 | 66.1 ± .5 | 67.3 ± .4 | 67.0 ± .4 | **68.1 ± .2** | **68.2 ± .1** | 40.8 ± .3 | 41.4 ± .3 | 43.2 ± .3 |
| | SVHN | 70.3 ± .3 | 69.9 ± .4 | 70.5 ± .4 | **70.8 ± .2** | 69.2 ± .4 | **70.7 ± .1** | **70.8 ± .1** | 30.3 ± .4 | 31.3 ± .3 | 34.3 ± .2 |
| | SYN | 88.2 ± .3 | 88.4 ± .3 | 89.8 ± .3 | 90.3 ± .2 | 86.6 ± .4 | 90.3 ± .2 | **90.6 ± .2** | 58.7 ± .4 | 62.3 ± .2 | 64.5 ± .3 |
| | Avg. | 79.8 ± .3 | 79.8 ± .3 | 80.8 ± .4 | 81.5 ± .2 | 79.9 ± .4 | **81.7 ± .2** | **81.8 ± .1** | 50.4 ± .3 | 51.8 ± .2 | 54.1 ± .3 |
| **NICO++** | Autumn | 82.9 ± .1 | 84.9 ± .2 | 84.8 ± .4 | 85.1 ± .1 | 86.3 ± .2 | 85.6 ± .2 | 86.0 ± .2 | 85.0 ± .3 | 85.9 ± .3 | **86.9 ± .4** |
| | Dim | 75.8 ± .1 | 78.0 ± .3 | 78.3 ± .2 | 79.3 ± .2 | 81.1 ± .2 | 80.4 ± .1 | 80.9 ± .2 | 78.5 ± .4 | 80.9 ± .3 | **81.6 ± .2** |
| | Grass | 84.9 ± .3 | 86.8 ± .1 | 86.5 ± .2 | 87.0 ± .3 | 87.8 ± .3 | 87.5 ± .3 | 87.7 ± .1 | 85.8 ± .3 | 87.2 ± .3 | **88.3 ± .3** |
| | Outdoor | 82.4 ± .1 | 84.3 ± .4 | 83.3 ± .2 | 84.8 ± .3 | 85.3 ± .1 | 85.3 ± .2 | 85.1 ± .2 | 83.4 ± .4 | 84.9 ± .2 | **85.7 ± .4** |
| | Rock | 83.9 ± .4 | 85.7 ± .2 | 85.3 ± .1 | 85.9 ± .1 | 87.0 ± .1 | 86.4 ± .3 | 87.0 ± .2 | 84.4 ± .4 | 85.8 ± .3 | **87.4 ± .4** |
| | Water | 77.5 ± .2 | 79.7 ± .2 | 78.8 ± .2 | 79.7 ± .1 | 80.3 ± .2 | 80.3 ± .1 | 80.1 ± .2 | 78.8 ± .4 | 80.2 ± .3 | **81.9 ± .4** |
| | Avg. | 81.2 ± .2 | 83.2 ± .2 | 82.8 ± .2 | 83.7 ± .2 | 84.6 ± .2 | 84.3 ± .2 | 84.5 ± .2 | 82.6 ± .4 | 84.2 ± .3 | **85.3 ± .3** |
| **DomainNet** | Clipart | 63.4 ± .2 | 59.4 ± .2 | 61.3 ± .2 | 63.9 ± .2 | 63.5 ± .2 | 64.2 ± .2 | 63.9 ± .2 | 63.9 ± .2 | 64.3 ± .1 | **64.8 ± .2** |
| | Infograph | 25.8 ± .3 | 25.0 ± .1 | 27.9 ± .2 | 29.7 ± .1 | 27.5 ± .3 | 30.8 ± .2 | 31.0 ± .2 | 34.9 ± .2 | 35.0 ± .2 | **36.7 ± .2** |
| | Painting | 49.7 ± .2 | 49.2 ± .2 | 51.4 ± .2 | 54.2 ± .1 | 53.1 ± .1 | 54.6 ± .2 | 55.2 ± .2 | 56.3 ± .2 | 57.2 ± .1 | **60.2 ± .2** |
| | Quickdraw | 11.8 ± .3 | 9.1 ± .2 | 10.1 ± .1 | 11.7 ± .2 | 10.9 ± .1 | **12.3 ± .2** | 12.9 ± .2 | 10.1 ± .3 | 10.8 ± .2 | **12.3 ± .3** |
| | Real | 61.6 ± .3 | 57.8 ± .1 | 61.0 ± .2 | 64.1 ± .2 | 63.4 ± .2 | 64.5 ± .2 | 64.3 ± .1 | 71.9 ± .3 | 72.6 ± .2 | **75.4 ± .3** |
| | Sketch | 48.1 ± .2 | 43.3 ± .1 | 50.6 ± .2 | 52.9 ± .2 | 52.1 ± .2 | 53.6 ± .2 | 53.2 ± .2 | 50.5 ± .2 | 52.4 ± .3 | **55.9 ± .4** |
| | Avg. | 43.4 ± .2 | 40.6 ± .2 | 43.7 ± .2 | 46.1 ± .2 | 45.1 ± .2 | 46.7 ± .2 | 46.8 ± .2 | 47.9 ± .2 | 48.7 ± .2 | **50.9 ± .2** |

Table 7: Leave-one-domain-out accuracies on VLCS, Terra Incognita, NICO++, DomainNet and Digits datasets for various knowledge distillation-based domain generalization approaches using different backbones.

| | | ResNet50 → ResNet18 | | | ViT-B/32 → ResNet18 | | | ResNet50 → ResNet50 | | |
|---|---|---|---|---|---|---|---|---|---|---|
| | | NKD | RISE | BOLD | NKD | RISE | BOLD | NKD | RISE | BOLD |
| **VLCS** | Caltech | 97.5 ± .3 | 96.6 ± .5 | **97.7 ± .3** | 97.5 ± .3 | 97.2 ± .3 | **97.7 ± .3** | 97.5 ± .2 | 96.9 ± .1 | **98.0 ± .2** |
| | LabelMe | 59.8 ± .6 | 60.9 ± .4 | **61.8 ± .4** | 60.4 ± .2 | 61.5 ± .3 | **63.0 ± .2** | 61.4 ± .5 | 62.4 ± .4 | **64.3 ± .2** |
| | Pascal | 77.4 ± .4 | 78.4 ± .4 | **78.9 ± .3** | 77.5 ± .2 | 79.5 ± .4 | **80.1 ± .2** | 79.2 ± .2 | 80.1 ± .3 | **80.9 ± .2** |
| | Sun | 68.3 ± .3 | 68.8 ± .5 | **69.3 ± .3** | 68.6 ± .3 | 69.5 ± .4 | **69.9 ± .3** | 70.1 ± .3 | 71.1 ± .1 | **71.4 ± .1** |
| | Avg. | 75.7 ± .4 | 76.2 ± .5 | **76.9 ± .3** | 76.0 ± .2 | 76.9 ± .3 | **77.7 ± .3** | 77.1 ± .3 | 77.6 ± .2 | **78.7 ± .2** |
| **Terra** | L38 | 23.5 ± .4 | 24.9 ± .4 | **31.3 ± .4** | 17.3 ± .3 | 18.6 ± .2 | **28.4 ± .2** | 28.1 ± .3 | 29.9 ± .3 | **41.8 ± .3** |
| | L43 | 29.2 ± .2 | 30.5 ± .2 | **39.5 ± .2** | 28.9 ± .2 | 29.9 ± .4 | **37.2 ± .3** | 32.9 ± .4 | 33.1 ± .2 | **37.2 ± .3** |
| | L46 | 24.8 ± .5 | 26.6 ± .4 | **30.2 ± .3** | 23.3 ± .4 | 24.8 ± .5 | **27.4 ± .4** | 31.1 ± .4 | 33.7 ± .3 | **27.4 ± .4** |
| | L100 | 10.8 ± .4 | 11.6 ± .2 | **16.6 ± .3** | 15.9 ± .4 | 16.2 ± .3 | **21.4 ± .3** | 17.0 ± .2 | 17.4 ± .3 | **21.4 ± .3** |
| | Avg. | 22.1 ± .4 | 23.4 ± .3 | **29.4 ± .3** | 21.4 ± .3 | 22.4 ± .4 | **28.6 ± .3** | 27.3 ± .3 | 28.5 ± .3 | **28.6 ± .3** |
| **NICO++** | Autumn | 78.6 ± .2 | 80.0 ± .3 | **80.7 ± .2** | 81.2 ± .3 | 82.0 ± .3 | **82.5 ± .2** | 83.4 ± .3 | 85.7 ± .3 | **86.2 ± .2** |
| | Dim | 70.2 ± .3 | 72.0 ± .3 | **72.6 ± .2** | 70.6 ± .3 | 72.4 ± .3 | **73.2 ± .2** | 77.1 ± .2 | 80.4 ± .2 | **80.9 ± .2** |
| | Grass | 80.5 ± .3 | 81.5 ± .2 | **82.1 ± .3** | 81.9 ± .3 | 82.8 ± .4 | **83.5 ± .3** | 85.2 ± .2 | 87.0 ± .3 | **87.7 ± .3** |
| | Outdoor | 78.5 ± .2 | 79.8 ± .4 | **80.6 ± .1** | 80.7 ± .3 | 82.2 ± .3 | **82.9 ± .3** | 82.5 ± .3 | 84.2 ± .2 | **85.0 ± .3** |
| | Rock | 79.0 ± .3 | 80.0 ± .4 | **80.6 ± .2** | 79.2 ± .4 | 80.6 ± .3 | **81.0 ± .2** | 83.7 ± .3 | 84.9 ± .4 | **85.6 ± .3** |
| | Water | 71.1 ± .3 | 71.5 ± .2 | **72.3 ± .2** | 71.6 ± .2 | 72.6 ± .3 | **73.3 ± .2** | 78.7 ± .3 | 79.6 ± .4 | **80.1 ± .4** |
| | Avg. | 76.3 ± .3 | 77.5 ± .3 | **78.1 ± .2** | 77.5 ± .3 | 78.8 ± .3 | **79.4 ± .2** | 81.8 ± .3 | 83.6 ± .3 | **84.3 ± .3** |
| **DomainNet** | Clipart | 40.3 ± .2 | 43.6 ± .2 | **43.8 ± .2** | 48.9 ± .2 | 51.6 ± .3 | **52.2 ± .4** | 50.7 ± .3 | 52.9 ± .2 | **53.3 ± .2** |
| | Infograph | 26.3 ± .2 | 29.2 ± .2 | **31.5 ± .3** | 26.8 ± .2 | 30.2 ± .3 | **32.2 ± .1** | 34.2 ± .3 | 34.7 ± .2 | **36.1 ± .2** |
| | Painting | 45.2 ± .2 | 47.3 ± .2 | **49.1 ± .2** | 47.8 ± .2 | 50.6 ± .3 | **52.4 ± .2** | 52.7 ± .3 | 55.5 ± .2 | **58.2 ± .2** |
| | Quickdraw | 5.1 ± .1 | 5.9 ± .1 | **6.4 ± .1** | 8.6 ± .2 | 9.2 ± .2 | **9.8 ± .2** | 6.2 ± .2 | 6.7 ± .3 | **7.1 ± .2** |
| | Real | 56.5 ± .2 | 60.9 ± .2 | **62.0 ± .1** | 60.3 ± .2 | 64.2 ± .2 | **65.9 ± .2** | 70.0 ± .2 | 71.8 ± .2 | **73.8 ± .2** |
| | Sketch | 41.3 ± .2 | 43.6 ± .2 | **44.4 ± .2** | 43.6 ± .3 | 45.4 ± .4 | **46.9 ± .3** | 46.2 ± .3 | 49.9 ± .3 | **52.6 ± .2** |
| | Avg. | 35.8 ± .2 | 38.4 ± .2 | **39.5 ± .2** | 39.4 ± .2 | 41.9 ± .3 | **43.2 ± .2** | 42.4 ± .2 | 45.2 ± .2 | **46.9 ± .2** |
| **Digits** | MNIST | 64.9 ± .2 | 67.0 ± .5 | **68.1 ± .3** | 58.7 ± .4 | 60.3 ± .4 | **63.8 ± .3** | 69.6 ± .4 | 70.5 ± .3 | **71.0 ± .2** |
| | MNIST-M | 30.0 ± .2 | 30.9 ± .4 | **32.1 ± .2** | 30.3 ± .3 | 31.6 ± .2 | **33.0 ± .3** | 31.5 ± .3 | 33.9 ± .2 | **36.0 ± .1** |
| | SVHN | 36.1 ± .3 | 38.3 ± .5 | **41.0 ± .2** | 21.7 ± .3 | 25.9 ± .4 | **28.4 ± .3** | 37.4 ± .5 | 39.6 ± .3 | **43.7 ± .2** |
| | SYN | 60.5 ± .3 | 62.6 ± .1 | **65.0 ± .2** | 54.3 ± .4 | 57.0 ± .4 | **59.9 ± .3** | 61.2 ± .3 | 63.9 ± .2 | **66.8 ± .3** |
| | Avg. | 47.9 ± .3 | 49.7 ± .4 | **51.6 ± .2** | 41.2 ± .3 | 43.7 ± .4 | **46.3 ± .3** | 49.9 ± .4 | 52.0 ± .3 | **54.4 ± .2** |

## A.3 IMBALANCED DATASET DISTRIBUTION

Figure 5 visualizes the dataset distributions for all benchmarks except Digits, which has a balanced distribution of 6,000 images per domain. Imbalanced dataset distribution is indeed a significant practical concern, particularly in ensuring the adequate training of domain experts. In our framework, domain experts are implemented as lightweight adapters consisting of a two-layer fully connected network rather than large-scale deep neural networks. This design allows these adapters to be effectively trained even in domains with only a few hundred images. Our experiments on benchmarks such as Terra Incognita, NICO++, and DomainNet demonstrate strong performance despite the presence of imbalanced distributions. For example, our framework achieves an average accuracy of 85.3% on NICO++ and 50.9% on DomainNet. While our approach does not explicitly address dataset imbalance, these results suggest that the framework is inherently robust to such challenges. In future work, we aim to extend the framework to more effectively address imbalance problems, tailoring it explicitly to such scenarios.

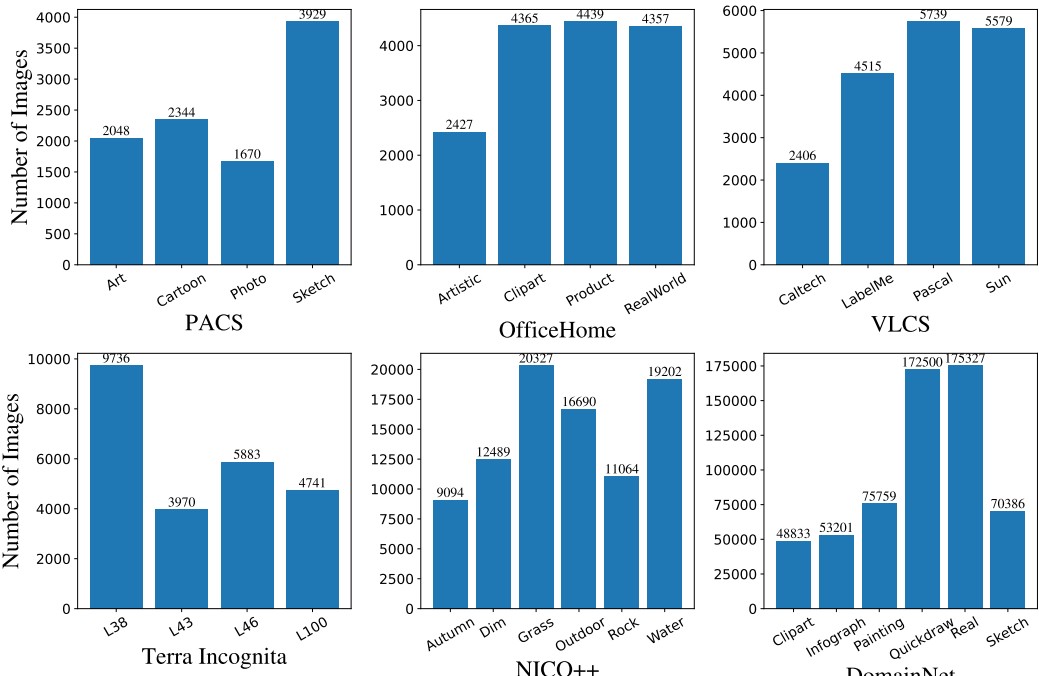

Figure 5: Dataset Distribution for PACS, OfficeHome, VLCS, Terra, NICO++, and DomainNet.

A.4 FURTHER EXPERIMENTS COMPARISON

Table 8 presents evaluation results across all benchmarks, comparing BOLD with CLIP-LinearProble (CLIP-LP) (Radford et al., 2021), CLIP-Adapter (CLIP-A) Gao et al. (2024), CoOp Zhou et al. (2022c), and CoCoOp (Zhou et al., 2022b). All experimental settings align with those discussed in the main paper. These evaluations were not included in the main paper because the listed baselines are CLIP-based fine-tuning methods. Compared to the baselines in the main paper, these methods utilize backbones with more parameters and are pretrained on larger datasets, making direct comparisons less fair. As shown in Table 8, despite having fewer parameters, BOLD achieves performance comparable to these CLIP-based fine-tuning methods across most benchmarks. Notably, on benchmarks like NICO++ and DomainNet, BOLD even outperforms these methods. This further highlights the effectiveness of BOLD in leveraging online knowledge distillation to integrate both domain-invariant and domain-specific knowledge.

Table 8: Leave-one-domain-out accuracies on VLCS, Terra Incognita, NICO++, DomainNet and Digits datasets for various CLIP-based fine-tuning approaches.

| | | CLIP-LP | CLIP-A | CoOp | CoCoOp | BOLD |
|---|---|---|---|---|---|---|
| **PACS** | Art | 88.6 ± .3 | 93.3 ± .1 | 89.4 ± .3 | 93.3 ± .5 | 88.1 ± .2 |
| | Cartoon | 89.6 ± .3 | 94.1 ± .1 | 92.7 ± .3 | 94.2 ± .1 | 86.9 ± .3 |
| | Photo | 98.5 ± .5 | 99.5 ± .0 | 98.8 ± .2 | 99.2 ± .3 | 97.9 ± .2 |
| | Sketch | 79.7 ± .3 | 82.2 ± .2 | 82.1 ± .4 | 78.8 ± .1 | 78.8 ± .3 |
| | Avg. | 89.1 ± .5 | 92.3 ± .1 | 90.7 ± .3 | 91.4 ± .4 | 87.9 ± .3 |
| **OfficeHome** | Artistic | 64.7 ± .4 | 70.5 ± .2 | 69.2 ± .3 | 71.4 ± .3 | 69.7 ± .2 |
| | Clipart | 51.7 ± .3 | 56.3 ± .1 | 54.8 ± .3 | 56.0 ± .2 | 58.9 ± .3 |
| | Product | 80.0 ± .1 | 83.1 ± .1 | 82.1 ± .0 | 84.6 ± .2 | 80.1 ± .2 |
| | RealWorld | 80.0 ± .1 | 83.8 ± .3 | 81.6 ± .1 | 83.6 ± .1 | 83.3 ± .2 |
| | Avg. | 69.1 ± .4 | 73.4 ± .2 | 71.9 ± .2 | 73.9 ± .2 | 73.0 ± .2 |
| **VLCS** | Caltech | 98.9 ± .3 | 100.0 ± .0 | 98.2 ± .3 | 100.0 ± .0 | 99.5 ± .2 |
| | LabelMe | 62.1 ± .4 | 60.0 ± .1 | 59.1 ± .2 | 69.6 ± .5 | 68.2 ± .3 |
| | Pascal | 83.2 ± .5 | 84.0 ± .2 | 80.0 ± .2 | 83.7 ± .3 | 83.5 ± .3 |
| | Sun | 76.3 ± .4 | 76.5 ± .2 | 73.1 ± .1 | 75.5 ± .3 | 71.7 ± .2 |
| | Avg. | 80.1 ± .5 | 80.1 ± .1 | 77.6 ± .2 | 82.2 ± .3 | 80.7 ± .2 |
| **Terra** | L38 | 35.3 ± .4 | 32.6 ± .5 | 34.0 ± .4 | 31.2 ± .3 | 38.3 ± .3 |
| | L43 | 46.2 ± .3 | 44.4 ± .1 | 44.9 ± .4 | 42.9 ± .3 | 41.2 ± .2 |
| | L46 | 32.1 ± .4 | 31.6 ± .3 | 33.7 ± .2 | 32.5 ± .3 | 31.5 ± .2 |
| | L100 | 27.5 ± .1 | 26.2 ± .3 | 28.5 ± .1 | 26.8 ± .4 | 28.1 ± .2 |
| | Avg. | 35.3 ± .3 | 33.7 ± .3 | 35.3 ± .3 | 33.4 ± .3 | 34.8 ± .2 |
| **Digits** | MNIST | 67.9 ± .5 | 63.7 ± .3 | 68.6 ± .4 | 61.3 ± .4 | 74.3 ± .3 |
| | MNIST-M | 48.9 ± .4 | 37.1 ± .3 | 48.1 ± .4 | 39.2 ± .4 | 43.2 ± .3 |
| | SVHN | 38.9 ± .3 | 32.8 ± .1 | 36.7 ± .4 | 34.7 ± .4 | 34.3 ± .2 |
| | SYN | 65.2 ± .2 | 66.4 ± .5 | 65.7 ± .3 | 64.9 ± .3 | 64.5 ± .3 |
| | Avg. | 55.2 ± .3 | 50.0 ± .3 | 54.8 ± .4 | 49.5 ± .4 | 54.1 ± .3 |
| **NICO++** | Autumn | 85.1 ± .1 | 85.2 ± .2 | 85.6 ± .1 | 85.9 ± .2 | 86.9 ± .4 |
| | Dim | 79.6 ± .2 | 80.8 ± .2 | 80.9 ± .2 | 81.3 ± .2 | 81.6 ± .2 |
| | Grass | 87.5 ± .3 | 88.1 ± .2 | 88.1 ± .2 | 88.6 ± .2 | 88.3 ± .3 |
| | Outdoor | 84.9 ± .2 | 85.6 ± .2 | 85.8 ± .1 | 86.7 ± .2 | 85.7 ± .4 |
| | Rock | 85.1 ± .1 | 86.1 ± .1 | 86.6 ± .1 | 87.0 ± .2 | 87.4 ± .4 |
| | Water | 78.0 ± .4 | 79.1 ± .3 | 79.7 ± .3 | 80.0 ± .3 | 81.9 ± .4 |
| | Avg. | 83.4 ± .2 | 84.2 ± .2 | 84.5 ± .2 | 84.9 ± .2 | 85.3 ± .3 |
| **DomainNet** | Clipart | 59.0 ± .1 | 59.2 ± .1 | 58.3 ± .2 | 60.0 ± .2 | 64.8 ± .2 |
| | Infograph | 31.9 ± .1 | 43.3 ± .2 | 41.1 ± .2 | 42.2 ± .1 | 36.7 ± .2 |
| | Painting | 49.2 ± .2 | 58.0 ± .1 | 56.5 ± .2 | 57.6 ± .2 | 60.2 ± .2 |
| | Quickdraw | 7.7 ± .1 | 8.5 ± .1 | 8.4 ± .1 | 8.3 ± .1 | 12.3 ± .3 |
| | Real | 70.2 ± .2 | 78.6 ± .1 | 74.9 ± .3 | 75.9 ± .2 | 75.4 ± .3 |
| | Sketch | 47.6 ± .1 | 52.9 ± .2 | 51.9 ± .2 | 50.7 ± .1 | 55.9 ± .4 |
| | Avg. | 44.3 ± .1 | 50.1 ± .1 | 48.5 ± .2 | 49.1 ± .1 | 50.9 ± .2 |

### A.5 Integrating Domain-Invariant and Domain-Specific Knowledge into One Model

Although the learning objectives of domain-invariant and domain-specific knowledge may appear conflicting, they can be complementary. The embedding space of the model is multi-dimensional, and not all dimensions need to serve both objectives simultaneously. By carefully designing the loss function, it is possible to coordinate the coexistence of these two types of knowledge within the embedding space (Sener & Koltun, 2018; Chen et al., 2024).

