# OpenReview forum: "Balancing Domain-Invariant and Domain-Specific Knowledge for Domain Generalization with Online Knowledge Distillation"
_ICLR.cc/2025/Conference — Submitted to ICLR 2025_

### Official Review · Reviewer_5xaq · 2024-10-19

**Soundness:** 3
**Presentation:** 3
**Contribution:** 2
**Rating:** 5
**Confidence:** 3

**Summary:**

The authors present an approach to improve the generalizability of deep learning models across multiple domains. It introduces the Balanced Online Knowledge Distillation (BOLD) framework, which leverages CLIP-based model as the teacher model and extract its domain-specific and domain-invariant knowledge through an online distillation strategy. Extensive experiments on multiple benchmarks indicate the effectiveness of the proposed method.

**Strengths:**

1. The writting and the structure of the paper is clear. The idea of online knowledge distillation is interesting and seems to be effective.
2. The authors provides theoretical analysis.
3. The experiments are thorough.

**Weaknesses:**

1. The proposed method is based on CLIP output adapters, which is an already well investigated topic in the previous works [1]. Even though the proposed online knowledge distillation ($L_{spc}$) seems to be novel, but it is only a tiny component of the proposed method. From this point of view, the contribution of this work to the community might be limited. The authors should further discuss this.
2. What is the intuition of using cross-entropy loss in the expert training? Why not just using the similarity-based metrics for text-image matching?
3. The implementation of the existing methods is unclear. For instance, for the baseline RISE, they claim to achieve around 90.2 with ResNet-50 backbone on PACS dataset. But in Table 1 their performance is only 86.6. The author should further clarify this. Besides, the authors mentioned that they used the setting "ViT-B/32 to ResNet50" for Table 1, while RISE used ResNet-50 as the backbone. The authors should describe more experimental details regarding this for a fair comparison.

[1] Gao, Peng, et al. "Clip-adapter: Better vision-language models with feature adapters." International Journal of Computer Vision 132.2 (2024): 581-595.

**Questions:**

Please refer to the Weaknesses.

---

> ### Author Response · Authors · 2024-11-21
>
> Dear Reviewer 5xaq, Thank you for your thoughtful feedback.
>
> 1. While the CLIP-Adapter serves as a parameter-efficient fine-tuning method to enhance CLIP’s performance on various downstream tasks, our methods use adapters to enable the teacher model to incorporate domain-specific knowledge from different domains. Our main focus is not on proposing new adapter structures, such as Tip-Adapter [1], but on addressing the challenge of distilling both domain-invariant and domain-specific knowledge into a single student model, which is a challenge in domain generalization. Our key contribution lies in the use of an online knowledge distillation strategy, which distinguishes it from existing approaches that primarily rely on offline distillation methods. By employing KL divergence, our method allows the teacher model (the domain expert) to dynamically update based on feedback from the student model, facilitating more effective knowledge transfer.
> 2. We will provide additional clarification in the revised version. Rather than relying solely on a similarity-based metric, we employ cross-entropy loss because it inherently includes calculating similarity metrics [2]. In CLIP, similarity is computed as a prerequisite for the cross-entropy calculation. For each domain d_i, we generate m prompts, where m is the number of class labels. Using cross-entropy loss allows us to not only maximize the similarity between an image and its ground truth prompt but also to minimize the similarity between the image and its unmatched class prompts. This dual objective aligns with the training strategy used in the original CLIP paper [2].
> 3. The results presented in Table 1 for NKD, RISE, and BOLD are based on knowledge distillation from ViT-B/32 to ResNet-50 for a consistent and fair comparison. This setup ensures the comparison is not limited to BOLD but applies uniformly across all methods. Table 2 provides additional results for NKD, RISE, and BOLD under different distillation settings, also ensuring fairness. Regarding the discrepancy in RISE’s performance, the original RISE implementation only includes four benchmarks (PACS, OfficeHome, VLCS, and Terra). To ensure fair comparison across all baselines and benchmarks, we replicated RISE in our library and conducted all experiments under the same conditions, including consistent data augmentation, prompts, and other settings. The code for our replication, along with the other baselines, is available in the GitHub repository linked in the paper.
>
> [1] Zhang, R., Zhang, W., Fang, R., Gao, P., Li, K., Dai, J., ... & Li, H. (2022, October). Tip-adapter: Training-free adaption of clip for few-shot classification. In European conference on computer vision (pp. 493-510). Cham: Springer Nature Switzerland.
>
> [2] Radford, A., Kim, J. W., Hallacy, C., Ramesh, A., Goh, G., Agarwal, S., ... & Sutskever, I. (2021, July). Learning transferable visual models from natural language supervision. In International conference on machine learning (pp. 8748-8763). PMLR.

---

> ### Author Response · Authors · 2024-11-25
>
> Dear Reviewer,
>
> As the discussion period draws to a close, we would greatly appreciate your feedback on whether our responses have sufficiently addressed your concerns. If we have successfully clarified or resolved the issues raised, we kindly ask you to consider revising your score.
>
> Thank you for your time and thoughtful consideration.
>
> Best regards

---

> ### Author Response · Authors · 2024-12-02
>
> Dear Reviewer 5xaq
>
> We hope this message finds you well. We sincerely appreciate your time and effort in reviewing our submission and providing valuable insights.
>
> We wanted to kindly follow up regarding our rebuttal as the discussion phase is nearing its conclusion.
>
> We would greatly appreciate any additional comments or feedback you may have regarding our response to your reviews. Your input is invaluable in clarifying and strengthening the work.
>
> If we have successfully clarified or resolved the issues raised, we kindly ask you to consider revising your score.
>
> Thank you once again for your time and consideration. We look forward to any further thoughts you might have.
>
> Best regards

---

> ### Comment · Reviewer_5xaq · 2024-12-03
> **Official Comment by Reviewer 5xaq**
>
> Thanks the authors for the response. After reading the rebuttal, my concerns regarding the performance divergence still remains. This point was also raised by Reviewer QY7Z. Therefore, I would like to maintain my original score.

---

### Official Review · Reviewer_Jua5 · 2024-11-02

**Soundness:** 2
**Presentation:** 3
**Contribution:** 3
**Rating:** 5
**Confidence:** 4

**Summary:**

Current knowledge distillation-based domain generalization approaches overlook the importance of domain-specific knowledge and rely on a two-stage training process, which limits the effectiveness of knowledge transfer. To overcome these limitations, this paper proposes the Balanced Online knowLedge Distillation (BOLD) framework for domain generalization, exploring the domain invariant for effective knowledge transfer while domain-specific knowledge is reserved. Experiments demonstrates its effectiveness.

**Strengths:**

1. The codes are provided, which makes it easy to reproduce performance.

2. The question raised is meaningful. How to retain domain-specific information is a point worth exploring in the field of knowledge transfer.

3. Theoretical proof is provided, and the effectiveness of the method is analyzed theoretically.

**Weaknesses:**

1. There is some deficiency in related works. The exploration of domain-specific has been reflected in some domain adaptation/domain generalization literatures in the past, and it needs to be reflected in related works. e.g., [1][2][3].

2. The variant of ablation study was a little too simple, and we expected to see the effect of domain-invariant and domain-specific knowledge separately and the corresponding analysis.

[1] Bui M H, Tran T, Tran A, et al. Exploiting domain-specific features to enhance domain generalization[J]. Advances in Neural Information Processing Systems, 2021, 34: 21189-21201.

[2] Seo S, Suh Y, Kim D, et al. Learning to optimize domain specific normalization for domain generalization[C]//Computer Vision–ECCV 2020: 16th European Conference, Glasgow, UK, August 23–28, 2020, Proceedings, Part XXII 16. Springer International Publishing, 2020: 68-83.

[3] Chang W G, You T, Seo S, et al. Domain-specific batch normalization for unsupervised domain adaptation[C]//Proceedings of the IEEE/CVF conference on Computer Vision and Pattern Recognition. 2019: 7354-7362.

**Questions:**

The student embedding is constrained by both the invariant distillation loss and the specific distillation loss. These two constraints aim to find cross-domain common information and domain-specific knowledge, respectively, which are potentially contradictory (orthogonal). Therefore, why can these two contradictory losses directly affect the same embedding and still ensure effectiveness? Intuitively, it seems impossible for one embedding to both share information across domains and be unique to each domain. I hope the author can address this concern.

---

> ### Author Response · Authors · 2024-11-21
>
> Dear Reviewer Jua5, Thank you for your thoughtful feedback.
>
> 1. We will incorporate the recommended literature into the Related Work section to offer a more comprehensive discussion.
> 2. Thank you for your feedback regarding the ablation study. To better address your concerns, could you kindly provide detailed requirements about any additional ablation experiments that you would like to include? In Table 3, we have already included results from an ablation study that examines (1) distilling invariant knowledge only, (2) distilling both invariant and specific knowledge in an offline setting, and (3) distilling both invariant and specific knowledge in an online setting. We would add a discussion in the revised version about how conducting an ablation study focusing solely on distilling specific knowledge does not align with the domain generalization context. As discussed in the paper, we aim to demonstrate how domain-specific knowledge can complement domain-invariant knowledge in practice. Relying exclusively on specific knowledge would inevitably result in poor performance, as it lacks the generalization capacity necessary for domain generalization tasks.
> 3. Thank you for raising this important question. We will provide additional clarification in Appendix A5 of the updated version. Although the learning objectives of domain-invariant and domain-specific knowledge may appear conflicting, they can be complementary. The embedding space of the model is multi-dimensional, and not all dimensions need to serve both objectives simultaneously. By carefully designing the loss function, it is possible to coordinate the coexistence of these two types of knowledge within the embedding space[1][2].
>
> [1]: Sener, O., & Koltun, V. (2018). Multi-task learning as multi-objective optimization. Advances in neural information processing systems, 31.
>
> [2]: Chen, L., Fernando, H., Ying, Y., & Chen, T. (2024). Three-way trade-off in multi-objective learning: Optimization, generalization and conflict-avoidance. Advances in Neural Information Processing Systems, 36.

---

> ### Author Response · Authors · 2024-11-25
>
> Dear Reviewer,
>
> As the discussion period draws to a close, we would greatly appreciate your feedback on whether our responses have sufficiently addressed your concerns. If we have successfully clarified or resolved the issues raised, we kindly ask you to consider revising your score.
>
> Thank you for your time and thoughtful consideration.
>
> Best regards

---

> ### Author Response · Authors · 2024-12-02
>
> Dear Reviewer Jua5
>
> We hope this message finds you well. We sincerely appreciate your time and effort in reviewing our submission and providing valuable insights.
>
> We wanted to kindly follow up regarding our rebuttal as the discussion phase is nearing its conclusion.
>
> We would greatly appreciate any additional comments or feedback you may have regarding our response to your reviews. Your input is invaluable in clarifying and strengthening the work.
>
> If we have successfully clarified or resolved the issues raised, we kindly ask you to consider revising your score.
>
> Thank you once again for your time and consideration. We look forward to any further thoughts you might have.
>
> Best regards

---

> > ### Comment · Reviewer_Jua5 · 2024-12-02
> >
> > I appreciate the author's response. My concerns regarding feature constraints still remain. According to the provided information, the model design includes two loss functions intended to ensure that the embedding vectors satisfy mutually orthogonal constraints. My primary concern is that these two constraints have completely opposing objectives, which raises questions about how the same feature can meet these seemingly contradictory requirements without specific design.
> >
> > Theoretically speaking, when two constraints are orthogonal to each other, it means that we expect the feature vectors in the embedding space to be independent in some dimensions, yet correlated in others. Without a clear mechanism or special design to ensure that this orthogonality and correlation can coexist harmoniously, standard feature extraction methods may struggle to achieve such effects. Particularly in complex datasets and model architectures, unadjusted features may degrade performance due to the interaction of loss functions during the optimization process. But seems like there is no special design in the paper to address this problem.

---

> > > ### Author Response · Authors · 2024-12-03
> > >
> > > Dear Reviewer Jua5
> > >
> > > Thank you for your thoughtful follow-up feedback.
> > >
> > > As referenced in [1], the two orthogonal constraints can be addressed from the perspective of Pareto Optimization. While the two loss functions may conflict in the feature space, one aiming to reduce cross-domain differences and the other seeking to enhance inter-domain differences, the objective of Pareto Optimization is to find a "Pareto optimal solution." This solution balances the two objectives, optimizing one without significantly compromising the other, which is common in multi-task learning.
> > >
> > > In addition to the theoretical rationale based on Pareto Optimization, we conducted extensive ablation studies to empirically validate the proposed framework. Our results demonstrate that optimizing both the invariant-distillation and specific-distillation losses further improves generalization performance compared to solely optimizing the invariant-distillation loss. These results are presented in Table 3, Section 4.2.
> > >
> > > [1]: Sener, O., & Koltun, V. (2018). Multi-task learning as multi-objective optimization. Advances in neural information processing systems, 31.

---

### Official Review · Reviewer_QY7Z · 2024-11-03

**Soundness:** 3
**Presentation:** 4
**Contribution:** 2
**Rating:** 5
**Confidence:** 4

**Summary:**

The paper propose the Balanced Online knowLedge Distillation (BOLD) framework for domain generalization, which employs a multi-domain expert teacher model, with each expert specializing in specific source domains to preserve domain-specific knowledge. This is the first investigation into the effectiveness of online knowledge distillation for domain generalization. The study also demonstrates that distilling both domain-invariant and domain-specific knowledge, rather than only domain-invariant knowledge, enhances model generalizability. Extensive experiments across seven domain generalization benchmarks validate the effectiveness of the proposed BOLD framework compared to state-of-the-art methods.

**Strengths:**

1. The writing of this work is good, and the explanation of the method is clear and easy to understand.
2. The authors provided theoretical analysis to support the proposed method.

**Weaknesses:**

1. The novelty is limited. In fact, learning both domain-invariant and domain-specific features is a common approach in domain generalization, with a lot of theoretical and experimental research already existing on it (e.g. Bui, Manh-Ha, et al. "Exploiting domain-specific features to enhance domain generalization." Advances in Neural Information Processing Systems 34 (2021)). The main difference between the author's work and previous research lies in the integration of this idea with knowledge distillation and different loss function design.

**Questions:**

1. As described by the authors, the results in Table 1 are based on leave-one-out evaluation strategy, with distillation from ViT-B/32 to ResNet-50. However, the evaluation results on some datasets are lower than those reported in the paper "In Search of Lost Domain Generalization"(e.g. ERM on PACS, 80.8 vs 83.3; ERM on VLCS, 75.5 vs 76.8), which also uses leave-one-out evaluation strategy and ResNet-50. The authors should explain these discrepancies and differences in experimental setup or implementation that might account for them.

---

> ### Author Response · Authors · 2024-11-21
>
> Dear Reviewer QY7Z, Thank you for your thoughtful feedback.
>
> 1. There are two key differences between our work and [1]. First, the motivation behind leveraging domain-specific knowledge differs significantly. In [1], the assumption is that there is a correlation between the domain-specific representation and the class label Y. In contrast, our work does not rely on this assumption. Second, our work introduces the novel approach of distilling both domain-invariant and domain-specific knowledge from a large pretrained model to a student model through a carefully designed online knowledge distillation strategy. By comparison, [1] addresses the problem by enabling the model to learn domain-invariant and domain-specific knowledge using an adversarial network with a meta-learning strategy. While both our work and [1] aim to address the challenge of integrating domain-invariant and domain-specific knowledge, they adopt distinct methodologies and pose different research questions, contributing unique perspectives to the field.
> 2. The performance differences arise from variations in the implementation protocols of baseline methods in domain generalization. In this field, two popular libraries are commonly used as implementation protocols: DomainBed [2] and DDAIG [3, 4]. By following the survey [5], our work follows the DDAIG protocol, which may result in discrepancies in the DomainBed performance of some older baselines, such as ERM, on classic benchmarks like PACS and VLCS. Importantly, all results in our work are presented under a consistent and fair comparison framework. Details of the implementation, including data loader, models, and benchmarks, are provided in the GitHub repository linked in the paper. Lastly, while these discrepancies affect specific baselines on classic benchmarks, they do not impact the conclusions drawn in our paper.
>
> [1] Bui, M. H., Tran, T., Tran, A., & Phung, D. (2021). Exploiting domain-specific features to enhance domain generalization. Advances in Neural Information Processing Systems, 34, 21189-21201.
>
> [2] Gulrajani, I., & Lopez-Paz, D. (2021). In search of lost domain generalization. ICLR.
>
> [3] Zhou, K., Yang, Y., Hospedales, T., & Xiang, T. (2020, April). Deep domain-adversarial image generation for domain generalization. In Proceedings of the AAAI conference on artificial intelligence (Vol. 34, No. 07, pp. 13025-13032).
>
> [4] Zhou, K., Yang, Y., Qiao, Y., & Xiang, T. (2021). Domain generalization with mixstyle. ICLR.
>
> [5] Zhou, K., Liu, Z., Qiao, Y., Xiang, T., & Loy, C. C. (2022). Domain generalization: A survey. IEEE Transactions on Pattern Analysis and Machine Intelligence, 45(4), 4396-4415.

---

> ### Author Response · Authors · 2024-11-25
>
> Dear Reviewer,
>
> As the discussion period draws to a close, we would greatly appreciate your feedback on whether our responses have sufficiently addressed your concerns. If we have successfully clarified or resolved the issues raised, we kindly ask you to consider revising your score.
>
> Thank you for your time and thoughtful consideration.
>
> Best regards

---

> ### Author Response · Authors · 2024-12-02
>
> Dear Reviewer QY7Z
>
> We hope this message finds you well. We sincerely appreciate your time and effort in reviewing our submission and providing valuable insights.
>
> We wanted to kindly follow up regarding our rebuttal as the discussion phase is nearing its conclusion.
>
> We would greatly appreciate any additional comments or feedback you may have regarding our response to your reviews. Your input is invaluable in clarifying and strengthening the work.
>
> If we have successfully clarified or resolved the issues raised, we kindly ask you to consider revising your score.
>
> Thank you once again for your time and consideration. We look forward to any further thoughts you might have.
>
> Best regards

---

### Official Review · Reviewer_XSvJ · 2024-11-04

**Soundness:** 3
**Presentation:** 3
**Contribution:** 3
**Rating:** 5
**Confidence:** 4

**Summary:**

This paper proposes a novel framework, termed Balanced Online Knowledge Distillation, for addressing the challenge of domain generalization. The paper initially underscores the critical role of domain-specific knowledge within the domain generalization task. Subsequently, it advocates for the integration of both domain-invariant and domain-specific knowledge through an online distillation strategy. Additionally, the paper endeavors to undertake a theoretical analysis to substantiate the efficacy of domain-specific knowledge in scenarios where the target domain exhibits resemblances to the source domain, while also elucidating the advantages of the online distillation strategy in enhancing generalization performance.

**Strengths:**

1.The paper proposes the Balanced Online Knowledge Distillation framework, which combines domain-invariant and domain-specific knowledge and employs the strategy of online knowledge distillation, which is a novel attempt.
2.The paper attempts to provide a theoretical analysis of the effectiveness of domain-specific knowledge in cases where the target domain has similar properties to the source domain, and how online knowledge distillation strategies can reduce the domain generalization error boundaries.
3.The English writing and essay organizations are good.

**Weaknesses:**

1.I would like to get more explanatory notes about the domain loss . From my perspective, the idea involves leveraging the text embedding of domain i as the positive sample of the Specific Embedding of domain i, while employing the text embedding of the domain other than i as the negative sample of the Specific Embedding of domain i. Given this interpretation, it may be more coherent to redefine the loss function.
2.When computing the Kullback-Leibler (KL) divergence loss, it is imperative to establish a clear delineation regarding which distribution is employed to guide the other. In this context, Equation (4) suggests that  signifies the utilization of the domain expert adapter to guide the student model. However, in Equation (6), the intended implication is for the student model to guide the domain expert adapter, a distinction not currently reflected in the discourse. Furthermore, the corresponding definition of  in Equation (5) remains unspecified.
3.Within the part labeled "Effectiveness of Domain-Specific Knowledge for Domain Generalization" in the Theoretical Discussion section, the current exposition primarily underscores the significance of shared attributes between the source and target domains. This emphasis appears somewhat detached from the core concept of "domain-specific knowledge."
4.Equation (7) does not seem to lead to the derivation of equation (11) within the specified conditions delineated in equation (10).
5.Does the student model learning from both Invariant Embedding and Specific Embedding create a conflict of knowledge?
6.While the results are advanced on multiple datasets, the performance on the Terra Incognita and Digits datasets is too poor.
7.Domain expert adapters should be a key module, but the corresponding details are missing from both the figure and the text.

**Questions:**

See Weaknesses.

---

> ### Author Response · Authors · 2024-11-21
>
> Dear Reviewer XSvj, Thank you for your thoughtful feedback.
>
> 1. Your understanding of the domain loss is accurate. As detailed in Equation 2, the domain loss assigns positive values to the corresponding domain (indicating minimization) and negative values to other domains (indicating maximization). This setup ensures the model learns to distinguish between the corresponding domain and other domains and aligns with common practices for loss function in similar contexts.
> 2. The KL divergence is indeed an asymmetric distance measure, and the direction of guidance - whether from the teacher to the student or from the student to the teacher differs when computing specific distillation loss. In the current version, we provide a general function for computing the KL divergence without explicitly clarifying this distinction. We will address this oversight in the updated version by clearly specifying the guidance direction in each context and providing the corresponding definitions.
> 3. In the theoretical discussion section, our analysis focuses on the shared domain-specific knowledge across the source and target domains, not on shared characteristics. The term “shared characteristics” is mentioned in the Introduction as a high-level description of domain-specific features. Then, the knowledge learned from these domain-specific features is what we call domain-specific knowledge. We appreciate your feedback and will ensure this distinction is made clear in the revised version.
> 4. In Equations (7) and (10), $L(h, D^i_s)$ represents the original student loss on source domain $i$ without guidance from the teacher model. Through the process of knowledge distillation, the student loss is reduced compared to the original loss $L(h, D^i_s)$, and it approximates $L(h_T, D^i_s) + \epsilon$. As a result, leveraging knowledge distillation allows us to replace $L(h, D^i_s)$ with $L(h_T, D^i_s)$. Since $L(h_T, D^i_s)$ is less or equal to $L(h, D^i_s)$, Equation (11) provides a tighter bound compared to Equation (7).
> 5. While incorporating domain-invariant and domain-specific knowledge may appear to introduce conflicting objectives, these forms of knowledge can also be complementary. The model’s embedding space is inherently multi-dimensional, allowing different dimensions to focus on distinct objectives without interference. By carefully designing the loss function, it is possible to harmonize these objectives, enabling their coexistence and effective coordination [1][2].
> 6. As discussed in the paper, the performance of knowledge-distillation-based methods is inherently influenced by the teacher model’s performance, a characteristic common to all such approaches. As shown in Table 4, the teacher model utilized in our work, CLIP, performs poorly on the Terra Incognita and Digits datasets. Consequently, methods like NKD, RISE, and BOLD also exhibit suboptimal performance on these datasets. However, by leveraging domain-specific knowledge and employing an online knowledge-distillation strategy, BOLD achieves significant improvements compared to other knowledge-distillation-based methods, demonstrating its relative effectiveness in addressing this limitation.
> 7. Thank you for highlighting this point. Initially, we did not include the adapter-based methods in our analysis, as doing so might have led to an unfair comparison given that CLIP-Adapter methods involve more parameters than classic RN50 methods. However, in response to your suggestion, we have added both the quantitative results and the qualitative results in Appendix A4 to provide a more comprehensive evaluation.
>
> [1] Sener, O., & Koltun, V. (2018). Multi-task learning as multi-objective optimization. Advances in neural information processing systems, 31.
>
> [2] Chen, L., Fernando, H., Ying, Y., & Chen, T. (2024). Three-way trade-off in multi-objective learning: Optimization, generalization and conflict-avoidance. Advances in Neural Information Processing Systems, 36.

---

> ### Author Response · Authors · 2024-11-25
>
> Dear Reviewer,
>
> As the discussion period draws to a close, we would greatly appreciate your feedback on whether our responses have sufficiently addressed your concerns. If we have successfully clarified or resolved the issues raised, we kindly ask you to consider revising your score.
>
> Thank you for your time and thoughtful consideration.
>
> Best regards

---

> ### Author Response · Authors · 2024-12-02
>
> Dear Reviewer XSvJ
>
> We hope this message finds you well. We sincerely appreciate your time and effort in reviewing our submission and providing valuable insights.
>
> We wanted to kindly follow up regarding our rebuttal as the discussion phase is nearing its conclusion.
>
> We would greatly appreciate any additional comments or feedback you may have regarding our response to your reviews. Your input is invaluable in clarifying and strengthening the work.
>
> If we have successfully clarified or resolved the issues raised, we kindly ask you to consider revising your score.
>
> Thank you once again for your time and consideration. We look forward to any further thoughts you might have.
>
> Best regards

---

### Official Review · Reviewer_de9i · 2024-11-08

**Soundness:** 2
**Presentation:** 3
**Contribution:** 2
**Rating:** 5
**Confidence:** 4

**Summary:**

The authors present an approach called Balanced Online Knowledge Distillation (BOLD) for domain generalization. BOLD uses a multi-domain expert teacher model to retain domain-specific knowledge and employs an online distillation strategy, allowing the teacher and student models to learn simultaneously. This setup enhances knowledge transfer and improves the model’s ability to generalize across unseen domains. Extensive experiments on seven benchmarks demonstrate the effectiveness of BOLD over state-of-the-art methods.

**Strengths:**

1. The clarity of the paper is commendable, and the Balanced Online Knowledge Distillation (BOLD) framework demonstrates effectiveness across seven benchmarks.

2. The authors provide both theoretical and empirical evidence to support the effectiveness of this method.

**Weaknesses:**

1. The theoretical analysis in this paper lacks rigor in establishing a strict upper bound for convergence and relies heavily on assumptions without concrete mathematical proofs. Specifically:
- Absence of Formal Proof for Upper Bound: The derivations, such as in Equations (11) and (13), introduce error terms  ϵ and ϵo ​but lack rigorous proof that these terms converge in the desired manner, resulting in an incomplete justification for the generalization bound.
- Reliance on Assumptions: The analysis assumes that incorporating domain-specific knowledge and applying online distillation will reduce domain discrepancy and empirical risk, yet these effects are only qualitatively described without mathematical substantiation or a precise convergence rate.
2.  The Balanced Online Knowledge Distillation (BOLD) framework is underexplored, as it does not account for the imbalanced dataset distribution across different domains. This limitation raises concerns about the assumption that all experts are well-pretrained.

**Questions:**

1. Can you provide a more rigorous theoretical analysis or formal proof for the convergence of the error terms introduced in Equations (11) and (13)?

2. How does the BOLD framework handle imbalanced dataset distributions across different domains?

---

> ### Author Response · Authors · 2024-11-21
>
> Dear Reviewer de9i, Thank you for your thoughtful feedback.
>
> 1. As the theoretical foundation for the effectiveness of online knowledge distillation remains underexplored in the broader field of knowledge distillation, our paper primarily focuses on providing a qualitative analysis. To address the limitations in theoretical rigor, we have conducted extensive experiments and ablation studies to empirically validate the effectiveness of the proposed method. We will provide more rigorous proof in the revised version.
> 2. Imbalanced dataset distribution is indeed an important practical concern, particularly regarding the adequate training of domain experts. In our framework, domain experts are lightweight adapters consisting of a two-layer fully connected network rather than large-scale, deep neural networks. This design ensures that these adapters can be adequately trained even for domains with only a few hundred images. As demonstrated in our experiments on benchmarks such as Terra Incognita, NICO++, and DomainNet, our method achieves strong performance despite the presence of imbalanced distributions. For instance, our framework achieves 85.3% average accuracy on NICO++ and 50.9% on DomainNet. Although our approach does not explicitly address dataset imbalance, these results suggest that the proposed framework is inherently robust to such challenges. To provide further clarity, we have expanded on this discussion and included a detailed visualization of dataset distributions in Appendix A3. In further work, we aim to extend our framework to better address pronounced imbalance problems, tailoring it more explicitly to such scenarios.

---

> ### Author Response · Authors · 2024-11-25
>
> Dear Reviewer,
>
> As the discussion period draws to a close, we would greatly appreciate your feedback on whether our responses have sufficiently addressed your concerns. If we have successfully clarified or resolved the issues raised, we kindly ask you to consider revising your score.
>
> Thank you for your time and thoughtful consideration.
>
> Best regards

---

> ### Author Response · Authors · 2024-12-02
>
> Dear Reviewer de9i
>
> We hope this message finds you well. We sincerely appreciate your time and effort in reviewing our submission and providing valuable insights.
>
> We wanted to kindly follow up regarding our rebuttal as the discussion phase is nearing its conclusion.
>
> We would greatly appreciate any additional comments or feedback you may have regarding our response to your reviews. Your input is invaluable in clarifying and strengthening the work.
>
> If we have successfully clarified or resolved the issues raised, we kindly ask you to consider revising your score.
>
> Thank you once again for your time and consideration. We look forward to any further thoughts you might have.
>
> Best regards

---

### Meta-Review · Area_Chair_mQgd · 2024-12-20

**Metareview:**

The paper received five reviews with the same rating score of 5. The reviewers raised several major concerns, including issues with the assumptions, the absence of concrete mathematical proofs for the theoretical analysis, lack of clarity regarding the model and loss terms, limited novelty and contribution, insufficient ablation studies, lack of implementation details, and inconsistent baseline performance. Despite the rebuttal, the reviewers remained unconvinced. Based on the overall feedback, this paper is recommended for rejection

**Additional Comments On Reviewer Discussion:**

Two reviewers engaged in discussions with the authors, but remained unconvinced by the authors' rebuttal.

---

### Decision · Program_Chairs · 2025-01-22

Reject